# Unifying the global phylogeny and environmental distribution of ammonia-oxidising archaea based on *amoA* genes

Ricardo J. Eloy Alves [1], Bui Quang Minh [2,4], Tim Urich[3], Arndt von Haeseler[2] & Christa Schleper[1]

Ammonia-oxidising archaea (AOA) are ubiquitous and abundant in nature and play a major role in nitrogen cycling. AOA have been studied intensively based on the *amoA* gene (encoding ammonia monooxygenase subunit A), making it the most sequenced functional marker gene. Here, based on extensive phylogenetic and meta-data analyses of 33,378 curated archaeal *amoA* sequences, we define a highly resolved taxonomy and uncover global environmental patterns that challenge many earlier generalisations. Particularly, we show: (i) the global frequency of AOA is extremely uneven, with few clades dominating AOA diversity in most ecosystems; (ii) characterised AOA do not represent most predominant clades in nature, including soils and oceans; (iii) the functional role of the most prevalent environmental AOA clade remains unclear; and (iv) AOA harbour molecular signatures that possibly reflect phenotypic traits. Our work synthesises information from a decade of research and provides the first integrative framework to study AOA in a global context.

[1] Archaea Biology and Ecogenomics Division, Department of Ecogenomics and Systems Biology, University of Vienna, Althanstrasse 14, 1090 Vienna, Austria. [2] Center for Integrative Bioinformatics Vienna, Max F. Perutz Laboratories, University of Vienna, Medical University of Vienna, Campus Vienna Biocenter 5, Dr. Bohr Gasse 9, 1030 Vienna, Austria. [3] Institute of Microbiology, Ernst-Moritz-Arndt University, Felix-Hausdorff-Strasse 8, 17487 Greifswald, Germany. [4]Present address: Ecology and Evolution, Research School of Biology, Australian National University, 2601 Canberra, ACT, Australia. Correspondence and requests for materials should be addressed to C.S. (email: christa.schleper@univie.ac.at)

The discovery of ammonia-oxidising archaea (AOA)[1-3] represents a major milestone in our understanding of the global nitrogen cycle and biology of archaea[4-6]. AOA are ubiquitous and abundant in the environment and play a major role in nitrification, the conversion of ammonia to nitrate via nitrite[6]. Nitrification is a central process in nitrogen cycling and availability, but also in emissions of the greenhouse gas nitrous oxide, groundwater pollution and eutrophication[7,8]. Moreover, AOA are predicted to be major contributors to carbon fixation in the oceans[9], where they have also been implicated in methane production[10] and cobalamin availability[11].

AOA comprise a diverse group of organisms formally defined as class *Nitrososphaeria* of the phylum Thaumarchaeota[12,13], whose classification has been often blurred by incongruent nomenclatures and phylogenies[5,6,13-17]. The diversity and ecology of AOA have been intensively studied, often in relation to that of ammonia-oxidising bacteria (AOB), and has relied mainly on analysis of *amoA* genes[18], which encode subunit A of the key metabolic enzyme ammonia monooxygenase (AMO). Consequently, *amoA* became the second most frequently sequenced marker gene in microbial ecology after the 16S ribosomal RNA (rRNA) gene, with the archaeal orthologue comprising 56% of sequences available (~58,000 archaeal sequences as of November 2016, excluding short fragments from high-throughput sequencing). The archaeal AMO belongs to the superfamily of copper-dependent membrane-bound monooxygenases (CuMMOs), which also include bacterial AMOs and hydrocarbon monooxygenases, although their evolutionary relationships have remained ambiguous[19-21].

Hundreds of *amoA*-based studies have collectively shown that AOA diversity and abundance in nature depend on multiple factors and are strongly partitioned by ecosystem[22-27]. AOA are typically broadly divided between marine and terrestrial groups, but several smaller-scale patterns of niche specialisation and potential ecophysiological adaptation have also been proposed. For example, seawater AOA comprise clades associated with different depths, suggested to reflect adaptations to different ammonia concentrations[26] and light exposures[28]. Depth-dependent patterns were also observed in freshwater lakes[27], and several marine/estuarine sediment clades were also suggested, including a low-salinity clade[22,29]. In turn, AOA diversity in soils has been associated mainly with pH[30]. However, associations between ecological, functional and phylogenetic patterns of AOA have been difficult to relate between studies, and their global phylogenetic breadth and coherence have remained largely ambiguous. Importantly, these patterns were mainly inferred on subjective phylogenetic levels in different phylogenies and often based on polyphyletic or unsupported clades[24,26,27,29-32], being thus prone to generate inconsistent categorical assumptions. To circumvent these bottlenecks, comprehensive phylogenetic/taxonomic frameworks that allow integration of sequence data and associated information in a global context are essential, as are those available for 16S rRNA genes[14,16,17]. These resources are becoming particularly relevant with increasing usage of short PCR amplicons of protein-coding marker genes that are unsuitable for de novo phylogenetic analyses. Ideally, such reference frameworks should also allow hypothesis testing and pattern comparisons across different levels of phylogenetic/taxonomic resolution. Furthermore, it is essential to establish explicit relationships between cultivated and environmental organisms to reliably extrapolate biological observations and assess current knowledge gaps.

Here, we perform extensive phylogenetic analyses of the known AOA diversity based on *amoA* genes and define a highly resolved taxonomy that provides a global framework to characterise AOA systematically. Based on this framework, we analyse the global environmental distribution and molecular signatures of AOA, as well as the relationships between the predominant lineages in nature and characterised strains. Additionally, we re-assess the evolution of CuMMO enzymes to infer the possible evolutionary origin of the archaeal AMO.

## Results

**Global phylogeny and taxonomy of AOA.** We compiled a database of 33,378 aligned nearly full-length archaeal *amoA* sequences (591–594 bp), after extensive curation of 45,136 publicly available sequences (Supplementary Fig. 1a, b, Supplementary Data 1). Preliminary analyses indicated that the initial dataset included a large number of chimeras (artefacts generated by PCR), which precluded reconstruction of stable phylogenetic trees. Chimeras were thus thoroughly filtered primarily based on a newly constructed chimera-free reference database (Supplementary Fig. 1b, Supplementary Data 1), as previously available only for 16S rRNA genes[33]. A total of 1994 chimeras were removed, representing 5.6% of all sequences and 7.3% of unique sequences in the initial dataset. Remarkably, clustering at 96% sequence identity had disproportionally increased the chimera fraction in the unfiltered dataset, with the chimera-filtered dataset yielding only half (54.7%) as many operational taxonomic units (OTUs) (Supplementary Fig. 1a, b). Likewise, it has been shown that chimeras can comprise up to 40% of 16S rRNA gene libraries and databases, and that clustering increased this fraction[33-36].

Next, we reconstructed a highly resolved phylogeny of 1190 archaeal *amoA* OTUs representing 33,256 sequences in the curated database clustered at 96% sequence identity, after excluding rogue sequences (Fig. 1, Supplementary Data 1, 2; see Methods). The final phylogenetic tree was selected from 70 independent trees inferred with three different methods, after comparative topology tests and congruency analyses (Supplementary Fig. 1c). Based on this phylogeny, we defined a multilevel *amoA* taxonomy strictly according to branches with both ultrafast bootstrap[37] ≥95% and Shimodaira–Hasegawa-like approximate likelihood ratio test (SH-aLRT)[38] ≥85%, each corresponding to an estimated confidence level of 95%[37] (Supplementary Data 2). As shown by a comprehensive 16S–23S rRNA gene phylogeny, class *Nitrososphaeria*[13] encompasses all known AOA and represents a monophyletic lineage at the exclusion of other (putative) thaumarchaeal groups (Fig. 2, Supplementary Fig. 2), in contrast to some previous phylogenies[5,15]. Consistent with this, the *amoA* phylogeny comprises four basal lineages congruent with the taxonomic orders of class *Nitrososphaeria*[13], abbreviated here to: NC (*Ca.* Nitrosocaldales), NS (*Nitrososphaerales*), NT (*Ca.* Nitrosotaleales) and NP (*Nitrosopumilales*) (Figs. 1 and 2). A fifth smaller clade was designated NT/NP-*Incertae sedis* (NT/NP-IS), as it could neither be assigned to lineages NT nor NP, according to our criteria, although it was suggestively placed at the root of NP. Each order-level lineage was divided into several broad clades designated by Greek letters. On this level, the phylogeny was also congruent, or at least compatible, with the rRNA gene phylogeny, namely the monophyly of genera *Nitrosopumilus*, *Ca.* Nitrosoarchaeum and *Ca.* Cenarchaeum (Fig. 2). Only clade NP-ε (*Ca.* N. brevis CN25[39]) showed conflicting placement between *amoA* and rRNA gene phylogenies, possibly due to the vast diversity of surrounding clades absent from the latter. Few supported "super-clades" (e.g., "NP-αβγ", Fig. 1) were omitted from the taxonomy to minimise potentially redundant higher ranks. Further taxonomic ranks were represented by numbers (e.g., NP-α-2.2.2.1) and were assigned only to clades with more than three OTUs, whereas clusters of two OTUs were designated *incertae sedis* ("IS"; see Methods for other taxonomic criteria).

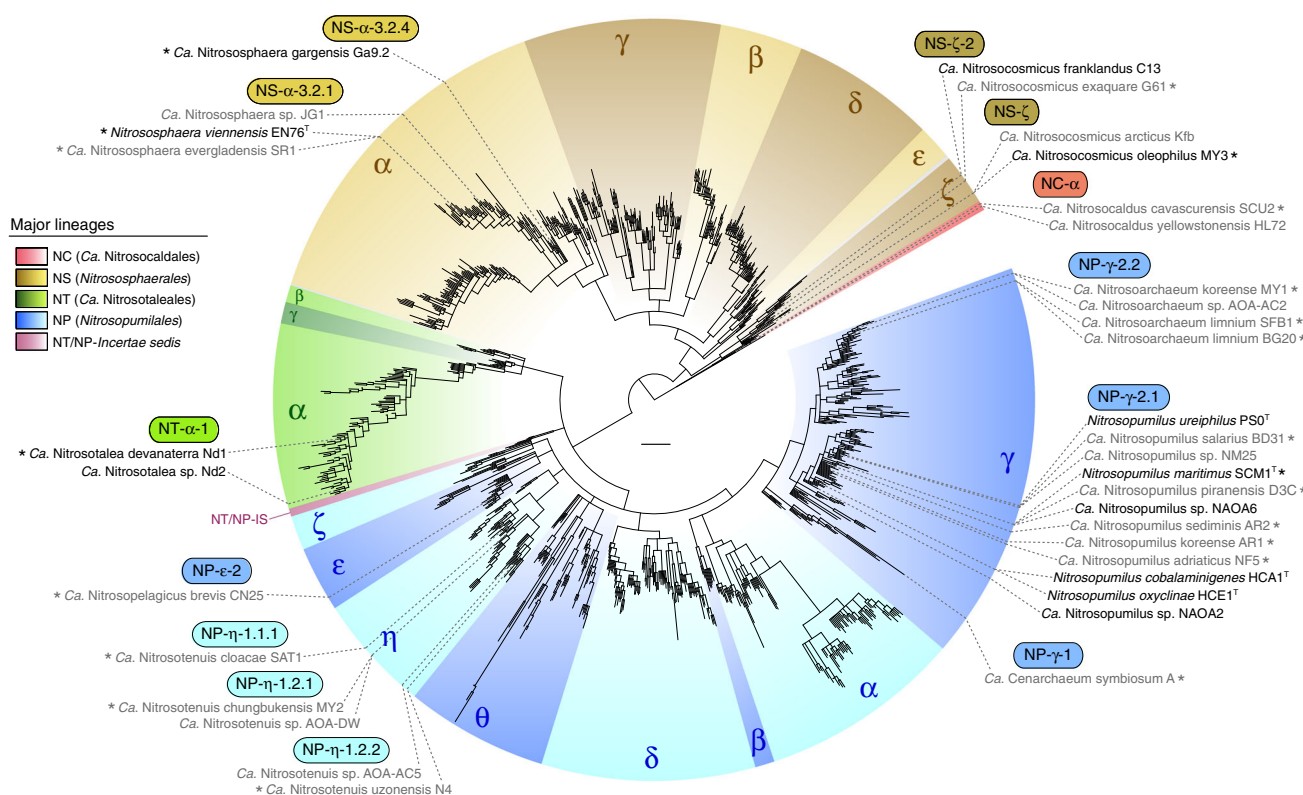

**Fig. 1** Global phylogeny of archaeal *amoA* genes from cultivated and environmental AOA. The phylogenetic tree was inferred from 1190 *amoA* sequences (591 aligned positions) representing 33,256 sequences in the curated database clustered at 96% sequence identity, after extensive phylogenetic analyses (see Supplementary Fig. 1). Order-level lineages are indicated by five main different colours, and major constituent subclades are indicated by Greek letters and different shades. Strains in pure or enrichment culture are indicated in black and grey text, respectively, and their abbreviated *amoA*-based classification is indicated in coloured boxes. Asterisks indicate organisms with sequenced genomes. The tree was inferred by maximum likelihood with IQ-TREE[80] using a GTR+F+I+Γ4 model, and branch support values were calculated from 1000 replicates. All branches shown have ultrafast bootstrap[37] ≥95% and SH-aLRT[38] ≥85%, each threshold corresponding to an estimated confidence level of 95%[37]; branches with lower support were collapsed and branch lengths were re-calculated with the GTR+F+I+Γ4 model. The scale bar represents 0.1 substitutions per nucleotide position. All clades were strictly defined based on the final tree topology according to the aforementioned criteria. Some original strain names were corrected to comply with International Code of Nomenclature of Prokaryotes, according to Kerou et al.[13] (see also Supplementary Table 1)

An editable phylogenetic tree is provided with taxonomic annotations compatible with FigTree (Rambaut, A., FigTree v1.4.3, http://beast.bio.ed.ac.uk/figtree, 2016) (Supplementary Data 2). The curated *amoA* database and taxonomy are implemented as reference databases for sequence classification using QIIME[40], mothur[41], LCAClassifier (CREST package[42]) and MEGAN[43]; tree and alignment files compatible with the Evolutionary Placement Algorithm[44] are also provided for sequence placement in the tree (Supplementary Data 3). The database is automatically available in the current version of LCAClassifier[42] (https://github.com/lanzen/CREST/releases/latest).

**Impact of chimeras on the archaeal *amoA* phylogeny.** The detrimental effect of chimeras on 16S rRNA gene diversity studies has been widely documented[35,36], but not their effect on diversity analyses of protein-coding genes, particularly on the phylogenies that typically guide them. To assess possible effects of chimeras on the *amoA* phylogeny, we inferred a phylogenetic tree from the 2203 unfiltered OTUs (≥96% sequence identity) following the same procedures used to infer the best tree (Fig. 1). The resulting tree showed drastic topological differences in relation to the chimera-free tree, regardless of branch support (Fig. 3): not only the monophyly of most major clades was lost, with constituent branches and subclades being often distantly placed, but also were relationships lost even between some order-level lineages, as

shown by the misplacements of lineage NT and part of clade NS-ε within lineage NP. Since chimeras were evenly distributed across the phylogeny, it seems unlikely that these incongruences resulted from a biased chimera filtering (Fig. 3).

**Uneven detection frequency of AOA.** AOA comprise a broad diversity of organisms that are greatly underrepresented by currently characterised strains and genomes (Figs. 1 and 2, Supplementary Table 1). Most of the 35 cultivated strains and over 46 genomes are affiliated with few lineages: for example, approximately half of all cultivated strains belong to clade NP-γ-2, representing the genera *Nitrosopumilus* and *Ca*. Nitrosoarchaeum. Although organisms from other frequently detected clades have been cultivated (NP-ε-2 and NP-η-1) (Figs. 1 and 4), clades encompassing >55% of all *amoA* genes in the database lack cultivated representatives, including the most abundant clade, NS-δ (Fig. 4, Supplementary Data 4). *AmoA* genes are distributed at almost even frequencies between lineages NS (49%) and NP (45%), with small fractions in NT (6%) and NC (<1%). However, the frequency of *amoA* genes is extremely uneven on lower phylogenetic levels, with half of all genes belonging to just three major clades: NS-δ (23%), NS-γ (13%) and NP-γ (15%) (Fig. 4, Supplementary Data 4). Such unevenness increases drastically at further lower levels, as reflected by the disproportional high frequency of few subclades and even specific OTUs, the latter

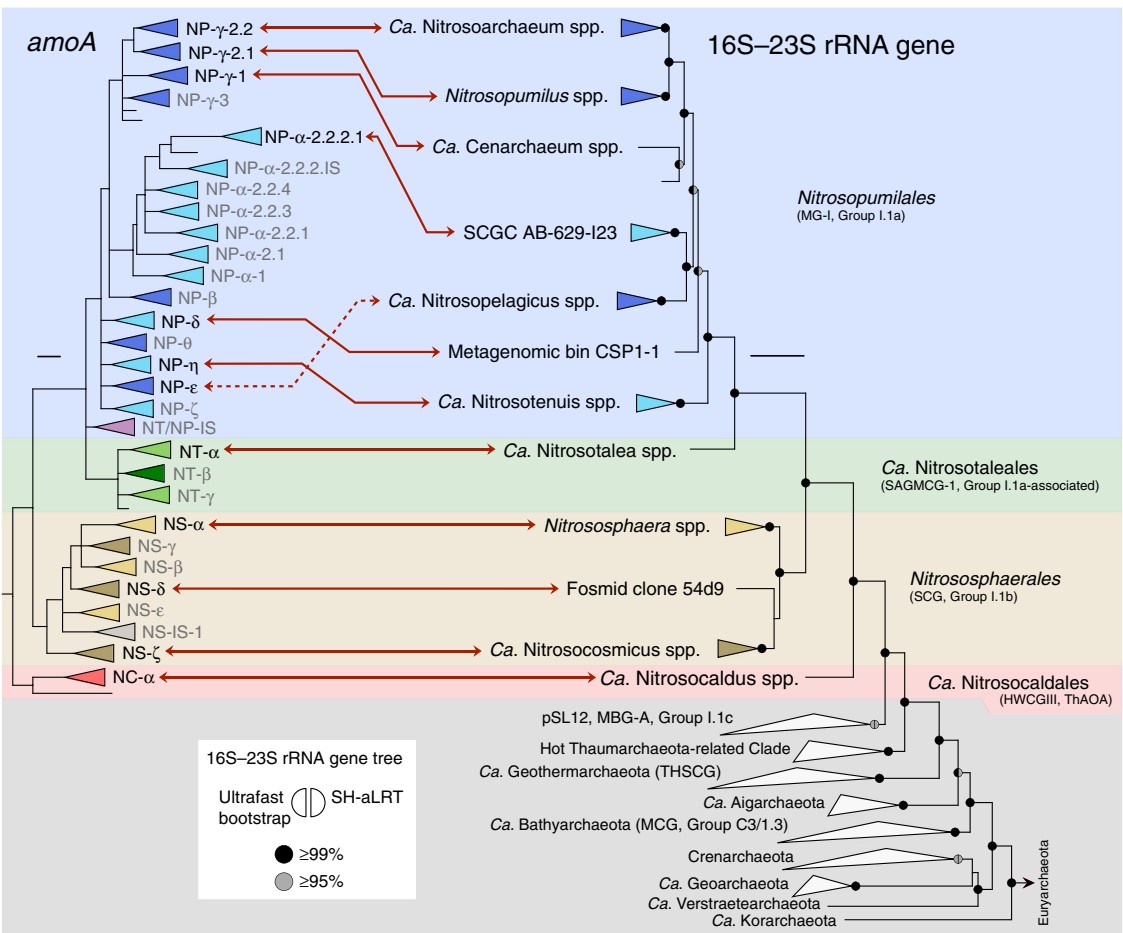

**Fig. 2** Congruency between *amoA* and concatenated 16S–23S rRNA gene phylogenies of AOA. The *amoA* tree represents a collapsed display of the tree in Fig. 1, and all branches shown have ultrafast bootstrap[37] ≥95% and SH-aLRT[38] ≥85%. The rRNA phylogenetic tree was inferred from 92 concatenated 16S and 23S rRNA gene sequences (4628 aligned positions), including representatives of all known Thaumarchaeota and TACK candidate phyla, all Crenarchaeota genera, and 8 Euryarchaeota classes (outgroup). Taxonomic orders of class *Nitrososphaeria* (i.e., AOA) are indicated by coloured backgrounds and the grey area represents other TACK archaea. Solid and dashed red arrows indicate corresponding *amoA* and rRNA gene lineages with congruent/compatible or conflicting topologies, respectively. Labels of *amoA* clades without known rRNA genes are shown in grey. An uncollapsed display of the rRNA gene tree is depicted in Supplementary Fig. 2; detailed AOA taxonomy and sequence accession numbers are provided in Supplementary Table 1. Names in brackets represent alternative nomenclatures in the literature. rRNA gene sequence alignments were based on archaea-specific structurally accurate seed alignments, and the tree was inferred by maximum likelihood with IQ-TREE[80] based on the GTR+F+R6 model with gene partitions using an edge-unlinked partition model[93]. Support values for branches with both ultrafast bootstrap[37] and SH-aLRT[38] >95% (1000 replicates) are indicated by semi-circles. Scale bars represent 0.1 substitutions per nucleotide position. MG-I Marine Group I, SAGMCG-1 South African Gold Mine Crenarchaeotic Group 1, SCG Soil Crenarchaeotic Group, HWCGIII Hot Water Crenarchaeotic Group III, ThAOA Thermophilic AOA, MBG-A Marine Benthic Group A, THSCG Terrestrial Hot Spring Crenarchaeotic Group, MCG Miscellaneous Crenarchaeotic Group

representing genes sharing ≥96% sequence identity (as reference, *Nitrosopumilus* species share 93–97% *amoA* gene sequence identity). This shows that, despite their broad phylogenetic diversity, environmental AOA are globally dominated by few highly abundant taxa, likely species or strains.

**Evidence of activity by few AOA clades.** To assess evidence of AOA activity beyond cultivated strains, we analysed information from 12 independent $^{13}C-CO_2$ stable isotope probing (SIP) studies of *amoA* genes with available sequence information, performed mainly in soils (Supplementary Table 2). Most AOA with evident autotrophic growth (i.e., dominant *amoA* genes in labelled SIP fractions) consistently derived from organisms closely related to strains cultivated as autotrophic ammonia oxidisers (Fig. 5). Although nitrification activity in SIP experiments was indirectly inferred, some studies provided the only current evidence of autotrophic growth by organisms from clades NS-β-1

and NP-α-2.2.4. Few genes from clades NS-γ, NS-δ and NS-ε were also detected in labelled SIP fractions, although it is unclear whether they represented technical artefacts or less abundant autotrophic organisms, as suggested by the low abundance of known autotrophic nitrifiers in some labelled SIP fractions (e.g., in NP-γ). On the other hand, independent studies suggested that particularly clades NS-δ[45] and NS-ε[46] might represent organisms that do not grow exclusively as autotrophic nitrifiers.

**Environmental diversity of AOA.** To characterise the global environmental distribution of AOA, we assigned all 33,378 *amoA* genes in our database to multiple manually curated environmental categories (i.e., habitats). To minimise ambiguous categorisations, only genes from oceanic regions were classified as marine, whereas those from coastal areas (e.g., bays) were grouped with genes from other marine–terrestrial interfaces as "estuarine-coastal"; similarly, genes from sediments from freshwater wetlands or waterlogged

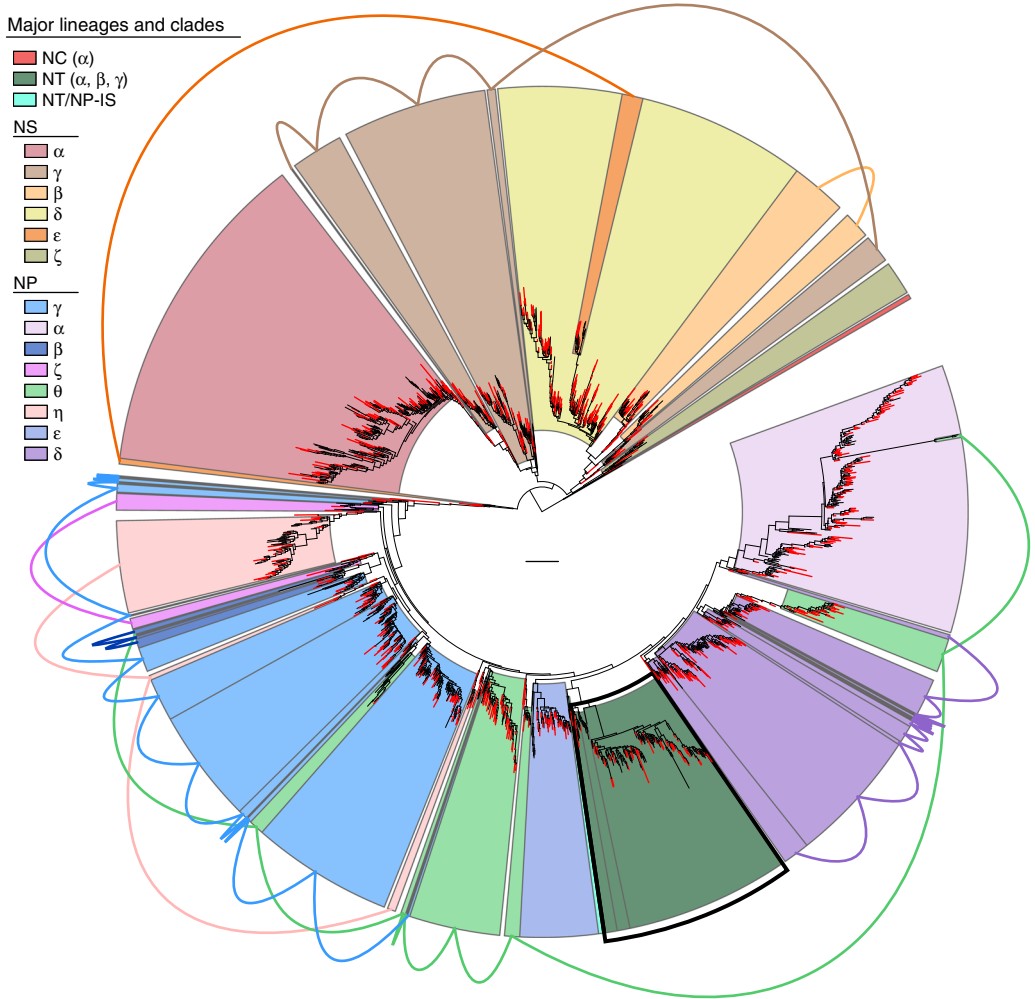

**Fig. 3** Effect of chimeras on the archaeal *amoA* phylogeny. The tree is based on 2203 sequences (591 aligned positions) representing our database prior to chimera filtering (35,372 sequences), clustered at 96% sequence identity. Major clades correspond to those defined based on the best tree inferred from the curated database (Fig. 1) and are colour coded as indicated in the figure. Framed areas with the same colour connected by lines indicate sequences from the same correct clade that are polyphyletic in this tree. Red branches represent chimeric sequences. The black-framed area highlights the misplacement of lineage NT (*Ca.* Nitrosotaleales) within lineage NP (*Nitrosopumilales*). The scale bar represents 0.2 substitutions per nucleotide position. The tree was calculated with IQ-TREE[80] following the same procedures used to obtain the best tree based on the curated database (see Supplementary Fig. 1)

areas (e.g., riparian areas and paddies) were grouped with those from soils as "soils-sediments" (see Methods for category definitions). The taxonomic distribution of *amoA* genes from each habitat is available in interactive Krona[47] charts (Supplementary Data 4-6).

The phylogenetic diversity of AOA is broadly segregated between soils-sediments and marine environments (Fig. 6, Supplementary Data 4), with 82% of AOA in soils-sediments belonging to lineage NS and 94% of marine AOA belonging to lineage NP. The 66% of soil-sediment AOA belong to just two clades that lack cultivated species: NS-δ (39%), represented by soil fosmid clone 54d9[2], and NS-γ (27%). The predominance of clade NS-δ in soils is consistent with that observed by a 16S rRNA gene survey of 146 soils from multiple ecosystems[48]. Additionally, clade NS-δ comprises considerable fractions of AOA in salt lakes (41%) and freshwater environments (26%) (Supplementary Data 4,5). Considering its global environmental prevalence, not only in soils, clade NS-δ either represents remarkably versatile organisms or widespread soil organisms that are frequently transferred to adjacent water bodies. AOA diversity in seawater and sediments is divided between nearly mutually exclusive clades

(Fig. 6, Supplementary Data 5): most seawater AOA belong to clades NP-ε-2 (54%) and NP-α-2.2.2.1 (35%), whereas sediments harbour greater diversity, mainly within clades NP-δ (32%), NP-γ (19%), NP-θ (17%) and NP-α (12%). The presence of multiple AOA clades in marine sediments has been previously reported, mainly based on 16S rRNA genes[49], but their relationship with most clades defined here could not be established (clades here are unrelated to previous clades, despite similar nomenclatures). Spurious AOA from lineage NS have also been found in marine sediments, although it is unclear whether they represented thriving organisms or dormant/dead cells transported from land. Similar to soil AOA, the predominant clades in the oceans lack cultivated representatives, with the exception of NP-ε-2, represented by *Ca.* N. brevis CN25[39]. Although organisms from clade NP-γ-2.1, particularly *N. maritimus* SCM1[T], have been extensively used as models for marine/oceanic environments[3,50,51], this clade comprises only 8% of all AOA identified in the oceans based on *amoA* genes (Supplementary Data 4). However, a study suggested that the abundance of organisms related to this clade has been underestimated in deep ocean waters due to primer biases[31]. Estuarine–coastal environments harbour the greatest

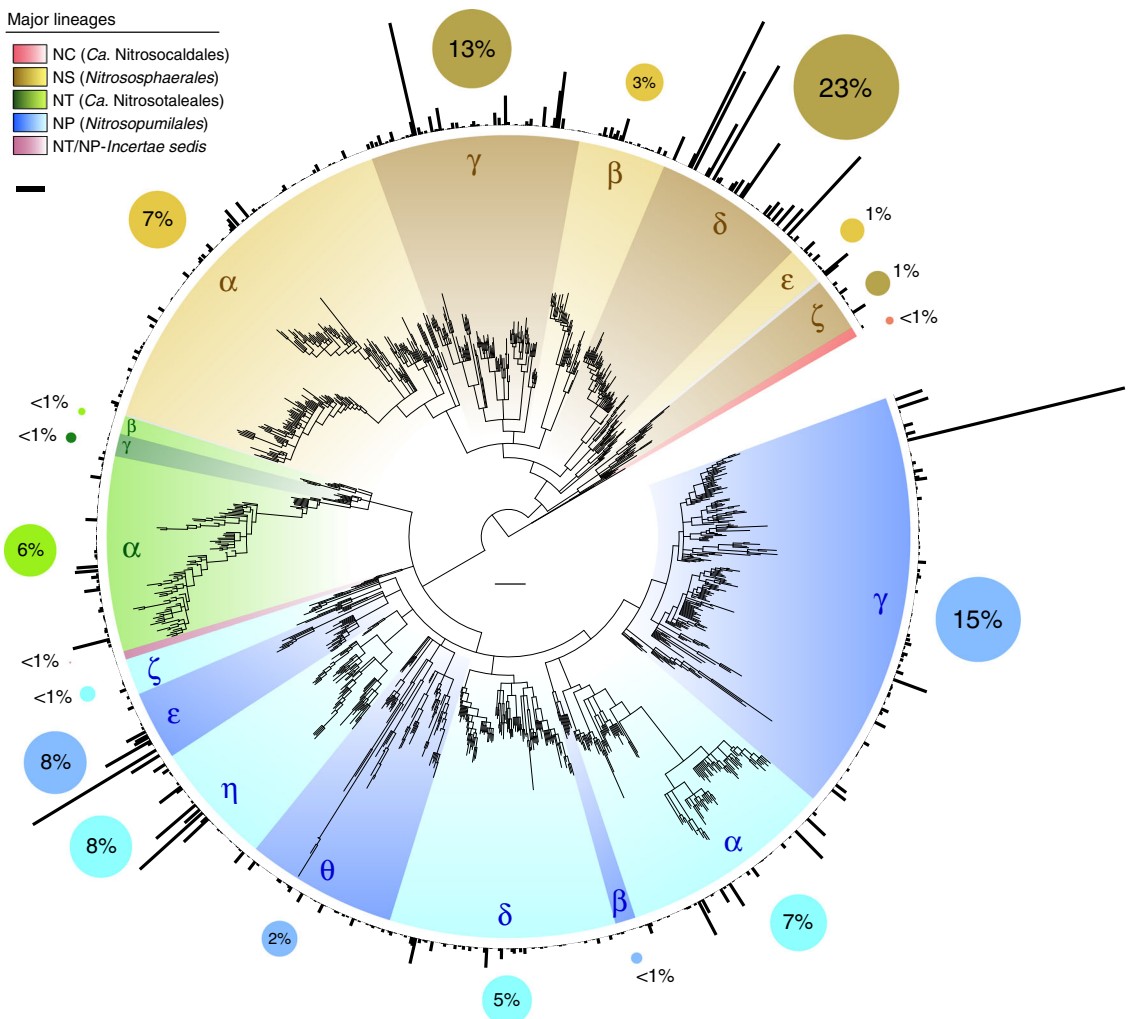

**Fig. 4** Phylogenetic distribution of archaeal *amoA* gene frequency. OTU and clade frequencies are based on all sequences in the curated database, including identical sequences (33,378 sequences). Black bars represent total sequence counts per OTU (96% sequence identity) and are drawn in proportion to the highest value (1960 sequences); the scale bar (below the colour legend) represents 250 sequences. Coloured circles show the fraction of sequences in the database associated with each major clade and their diameter is relative to those fractions. The tree scale bar represents 0.1 substitutions per nucleotide position. Sequence frequencies in all clades and subclades are also available in an interactive Krona[47] chart (Supplementary Data 4). See legend of Fig. 1 for tree description

phylogenetic diversity of AOA, spanning most major clades, possibly due to continuous inputs of organisms from marine, terrestrial and freshwater sources; nevertheless, 73% of estuarine–coastal AOA belong to lineage NP, and 37% are associated with clade NP-γ alone. Likewise, the diversity of AOA in freshwater environments greatly overlaps with that in soils-sediments, suggesting similar interchanges of organisms, despite the predominance of some clades in freshwater (e.g., NP-η, NP-γ and NS-δ) (Supplementary Data 4).

**Habitat specificity of AOA**. Based on the environmental categories defined here, the habitat specificity of AOA varies extensively across their global phylogenetic diversity and often contrasts with generalised assumptions in the literature (available in interactive Krona[47] charts in Supplementary Data 7-9). For instance, lineages NP and NS are commonly referred to as Marine Group I (MG-I) and Soil (Crenarchaeotic) Group (SCG), respectively[5,6,16,26,31,49], reinforcing the widespread notion that they are exclusively present in those environments. However, these lineages have a much broader environmental distribution:

only 45% of lineage NP was actually detected in unambiguously marine environments, whereas 68% of lineage NS was detected in soils and terrestrial sediments (Fig. 6, Supplementary Data 7). While clades NP-α, NP-ε, NP-θ and NP-ζ do occur mainly in marine environments, clade NP-γ occurs more frequently in estuarine–coastal and freshwater environments, whereas clade NP-η occurs mainly in freshwater and soils-sediments. Within lineage NS, NS-γ is the clade most specifically associated with soils-sediments (84%), while 29–59% of other clades were detected in aquatic ecosystems; for example, 59% of clade NS-ε and 47% of clade NS-β were found more often in freshwater, marine and/or estuarine–coastal environments. NT clades occur almost evenly in soils-sediments and freshwater, while lineage NC is nearly exclusive to terrestrial hot springs (Fig. 6, Supplementary Data 7).

Most of these broad AOA clades include also distinct, and often very specific, putative ecotypes (Fig. 6, Supplementary Data 7-9). For example, the cosmopolitan clade NP-γ comprises several subclades with distinct environmental distributions: NP-γ-1 was found mainly in marine aquaria and sediments; NP-γ-2.1 occurs mainly in estuarine–coastal environments, but includes

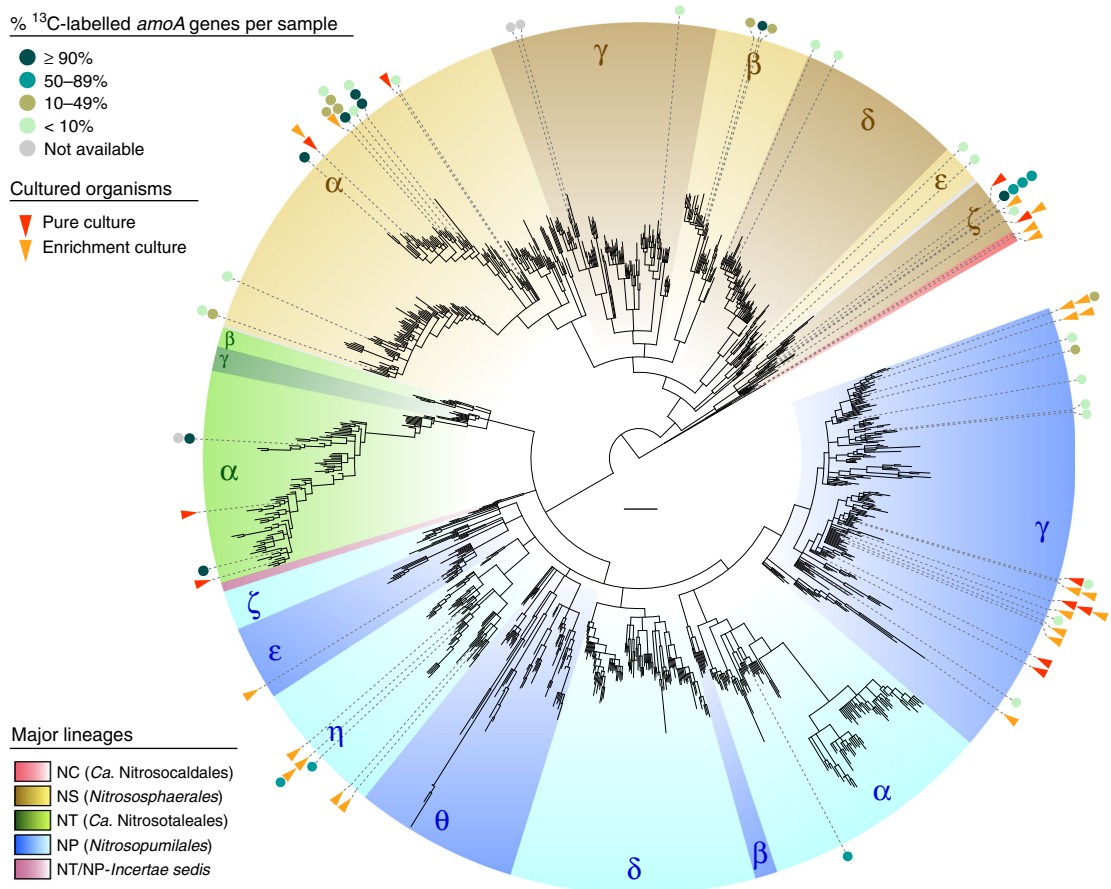

**Fig. 5** Phylogenetic distribution of evidence for autotrophic growth and ammonia oxidation among AOA. Phylogenetic placement of *amoA* genes of AOA in pure (red triangles) or enrichment culture (orange triangles) with $NH_3$ and $CO_2$ as sole energy and carbon sources, respectively, and representatives of $^{13}$C-labelled *amoA* genes from 12 published $^{13}$C-$CO_2$ stable isotope labelling (SIP) experiments (circles). All 44 sequences represented share ≥96% identity with their corresponding reference sequence in the tree. Filled circles indicate representative sequences and are coloured according to relative abundance of associated *amoA* genes among all $^{13}$C-labelled *amoA* genes obtained from the corresponding sample. The scale bar represents 0.1 substitutions per nucleotide position. References for cultivated strains and SIP studies included are shown in Supplementary Tables 1, 2. See legend of Fig. 1 for tree description

further subclades more frequent in marine water, sediments, aquaria or sponges; NP-γ-2.2 occurs mainly in estuarine–coastal and freshwater environments, but includes subclades often found in soils or marine sediments. Although "low-salinity" and "freshwater" clusters related to marine AOA from clade NP-γ have been suggested[27,29], their phylogenetic relationships were ambiguous, and thus this clade has remained broadly regarded as prototypically marine. We also identified distinct subclades independently associated with terrestrial hot springs and marine eukaryotes, regardless of the environmental occurrence of closely related organisms (Fig. 6, Supplementary Data 8). In addition to the thermophilic lineage NC, only clades NP-η-1.2.2 and NS-α-3.2.3.2 occur consistently in terrestrial hot springs, despite considerable, but spurious, AOA diversity in these environments[52]. In turn, several clades associated with marine eukaryotes, namely sponges and macroalgae, have emerged independently from broader clades prevalent in different marine environments, namely sediments (NP-θ-4 and NP-ζ-2), seawater (NP-ε-1) or aquaria (NP-γ-1.1). The consistent occurrence of most of these clades in both types of putative host (NP-θ-4, NP-ζ-2 and NP-ε-1) further supported that they represent true, but nonspecific, symbionts (Supplementary Data 9).

Furthermore, we re-assessed the distribution of AOA according to ocean water column depth and soil pH, which constitute the best-established factors of niche segregation among AOA[30,53].

Seawater AOA are typically divided between "shallow" (Water Column A) and "deep" (Water Column B) clades. However, we found that these presumed clades vary between studies, with the "shallow" clade corresponding to either clade NP-ε[53,54], NP-γ[28,55] or both[26,32], while the "deep" clade corresponds more generally to clade NP-α-2.2.2.1. Based on depth information from 2757 *amoA* genes, 64% of clade NP-ε was indeed detected above 200 m depth (i.e., epipelagic zone), with only 5% detected in deeper waters; however, the small fraction of clade NP-γ found in seawater (NP-γ-2.1.3.1) derived mainly from deep rather than shallow waters, as also observed by a recent study[31] (Fig. 7, Supplementary Data 9). Clade NP-α-2.2.2.1 was more evenly detected both above (33%) and below (55%) 200 m depth, indicating that it is not specifically a deep-water clade; nevertheless, this clade likely represents most thaumarchaea that comprise up to ~40% of picoplankton in deep waters[56].

Based on soil pH information from 4754 *amoA* genes, AOA in acidic soils (pH < 6.5) consistently represent few specific clades (lineage NT and few NS clades, particularly NS-γ), whereas those in neutral (pH = 6.5-7.5) or alkaline (pH > 7.5) soils have more diverse and sparser phylogenetic affiliation (Fig. 7, Supplementary Data 9). As previously observed[30], most soil AOA from lineage NT were detected in acidic soils, or at least with pH < 7.5, although this lineage is almost as common in freshwater (Supplementary Data 7, 9). Other AOA in acidic soils represent

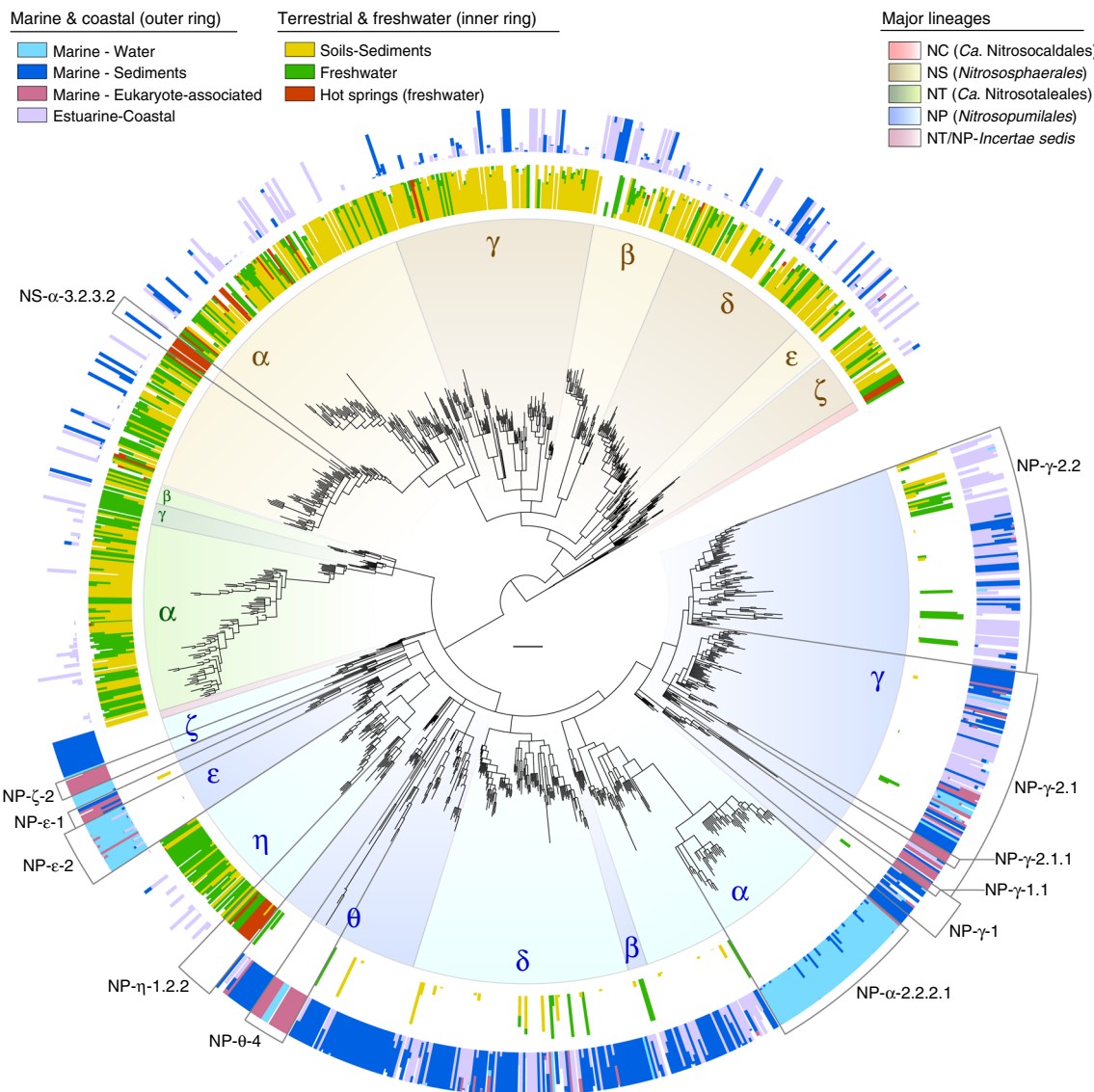

**Fig. 6** Distribution of archaeal *amoA* genes in five broad habitats. The environmental occurrence of 31,932 *amoA* genes unambiguously assigned to one of five broad habitats is represented as fraction of sequences in each OTU (96% sequence identity). Fractions were calculated in relation to total number of sequences in the OTU assigned to a habitat within the same category level, excluding those only assigned to the broader category above (e.g., fraction of sequences from each marine habitat were calculated relative to all marine sequences, excluding those that could only be categorised as marine). Subclades specified in the main text are indicated by grey lines. The scale bar represents 0.1 substitutions per nucleotide position. The absolute and relative phylogenetic/taxonomic distribution of sequences in these and more specific habitats are available in interactive Krona[47] charts (Supplementary Data 4-9). See legend of Fig. 1 for tree description

very specific low-pH ecotypes within broader clades widespread in non-acidic soils, namely NS-γ-2.2, NS-γ-2.3.2, NS-α-1, NS-α-2 and NS-ζ-2 (Fig. 7, Supplementary Data 9). Despite the suggestive recurrence of clade NS-δ in neutral soils, we could not identify clades specifically associated with neutral or alkaline soils, similar to those in acidic soils. In contrast to the general pH-driven evolution of soil AOA previously proposed[57], our results indicate that soil AOA include only distinct low-pH ecotypes, which apparently emerged from more cosmopolitan clades in relatively recent evolutionary events. In turn, other AOA clades found in soils occur over a wider soil pH range, and are also frequent in other environments where their relationship with pH is ambiguous.

**Distinct molecular signatures among AOA.** We analysed the phylogenetic distribution of DNA base composition and synonymous codon usage of 33,256 archaeal *amoA* genes to identify

possible molecular patterns associated with their global diversification (Fig. 8; interactive Krona[47] chart in Supplementary Data 10). Additionally, we compared these properties in all protein-coding genes from 35 genomes of AOA (hereafter referred to as "genomic") with those of their respective full-length *amoA* genes (Fig. 9) to determine if the latter can serve as general indicators of genomic traits of AOA.

Guanine–cytosine (GC) content of *amoA* genes varies between 38 and 58% (Fig. 8a), and closely mirrors variation in genomic GC content (Fig. 9a). This variation is mainly driven by an exceptional high-GC bias, relative to the average GC content, of clade NS-α and few small clades from lineage NP (Fig. 8a, Supplementary Data 10). GC content otherwise varies within a relatively narrow range and most other clades have GC content below average, except for a few subclades within NS-β, NS-γ and NP-η. The most prominent GC biases within lineage NP correspond to clades of putative symbionts of marine sponges

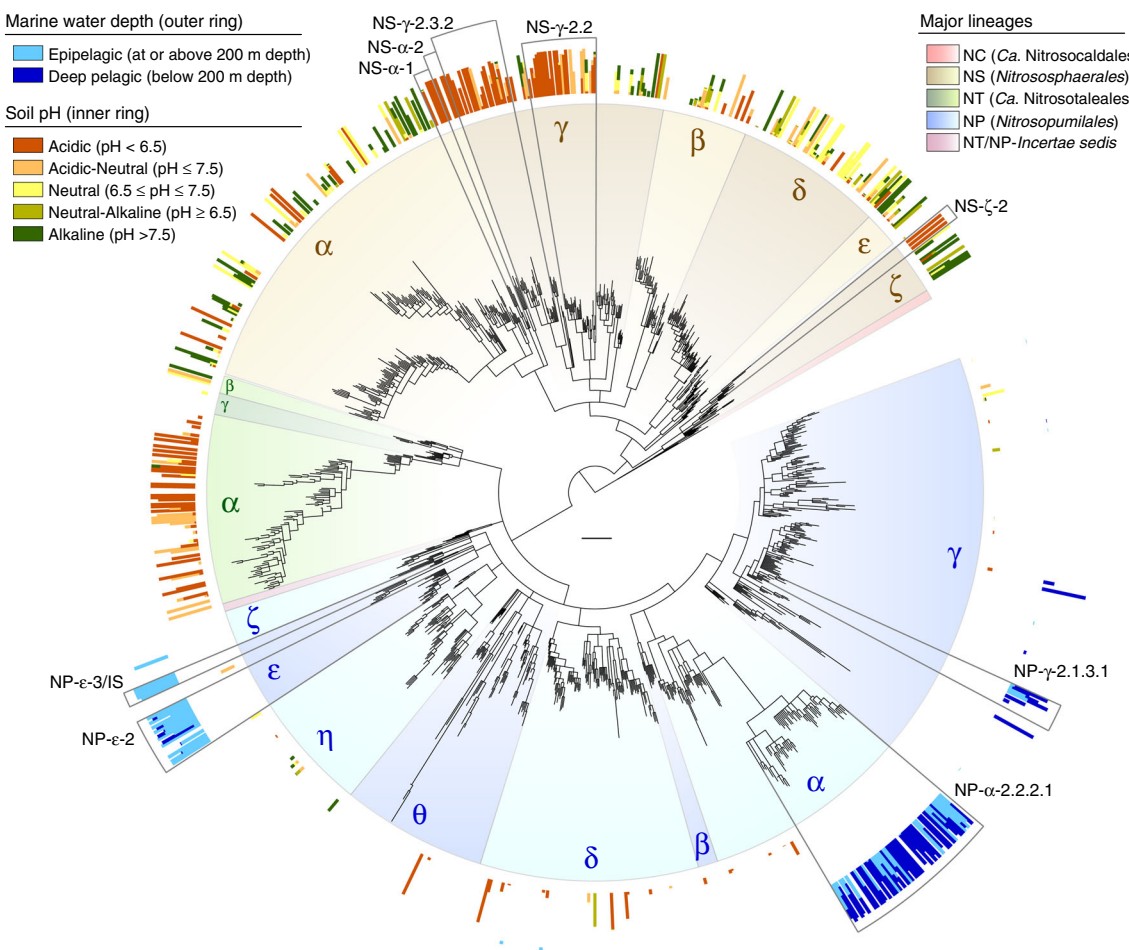

**Fig. 7** Phylogenetic distribution of archaeal *amoA* genes according to soil pH and marine water depth. The occurrence of *amoA* genes in distinct water column zones (2757 sequences) and soils with different pH (4754 sequences) is represented as fraction of sequences in each OTU (96% sequence identity). Fractions were calculated in relation to total number of sequences assigned to a niche/habitat within the same category level, and excluding those only assigned to the broader category above (e.g., fraction of each seawater depth among all sequences that could be assigned to a depth category). Subclades specified in the main text are indicated by grey lines. The scale bar represents 0.1 substitutions per nucleotide position. The absolute and relative phylogenetic/taxonomic distribution of sequences from these habitats are also available in interactive Krona[47] charts (Supplementary Data 6, 9). See legend of Fig. 1 for tree description

and macroalgae (Fig. 6, Supplementary Data 8), with the relatively high-GC content previously observed only in *Ca.* C. symbiosum A[12] rather constituting a common trait of most symbiont clades (NP-θ-4, NP-γ-1.1 and NP-γ-2.1.1) (Fig. 8a, Supplementary Data 10). This contrasts with the general low-GC trend of endosymbiotic bacteria[58], which was only consistent with the low-GC bias of symbiont clade NP-ε-1. Purine content of *amoA* genes varies within a narrower range than GC content (46–52%), but often shows greater variation between closely related clades (Fig. 8b); this variation, however, does not reflect that of the genomes (Fig. 9b). Purine content presumably plays a greater role in nucleic acid stability in low-GC organisms[58], and many *amoA* clades indeed show opposite trends in GC and purine contents (e.g., NP-γ, NP-η and NT-α). However, this is not always the case, indicating that variation in purine content among *amoA* genes does not result only from trade-offs with genomic GC content (Fig. 8a, b, Supplementary Data 10).

The effective number of codons (Nc) quantifies the usage of alternative synonymous codons, and varies between 20, when each amino acid is exclusively encoded by one codon (i.e., more biased), and 61, when amino acids are equally likely encoded by any possible codon[59]. Nc values of *amoA* genes varies across the whole possible range (Nc = 20–61) and show consistent trends

within some broad clades (e.g., NS-γ and NP-γ), but also opposite trends between closely related subclades (e.g., within NS-α and NP-δ) (Fig. 8c, Supplementary Data 10). Although the Nc of *amoA* genes is positively correlated with genomic Nc ($P < 0.0001$), the relatively low coefficient of determination (linear regression model; $R^2 = 0.569$) indicates that Nc patterns among *amoA* genes do not directly represent genomic codon biases (Fig. 9c). It has been shown that genomic codon usage biases can reflect the environmental distribution of an organism[60,61]: organisms with broader habitat range have generally lower Nc than those in restricted or stable environments, presumably reflecting a more efficient regulation of gene expression under different environmental conditions[61]. Some *amoA* clades show Nc patterns consistent with those trends, possibly reflecting difference in stringencies in regulation of *amoA* gene expression: for example, the cosmopolitan clade NP-γ has consistently more biased codon usage than average, whereas clades nearly exclusive to soils (NS-γ) or seawater (NP-α-2.2.2.1) have a much more relaxed codon usage (Fig. 8c, Supplementary Data 7, 8, 10).

We used the global codon adaptation index (gCAI)[62] to assess variation in synonymous codon usage (i.e., preferred codons) among *amoA* genes by comparing their codon usage with the dominant codon bias among all *amoA* genes. gCAI compares the

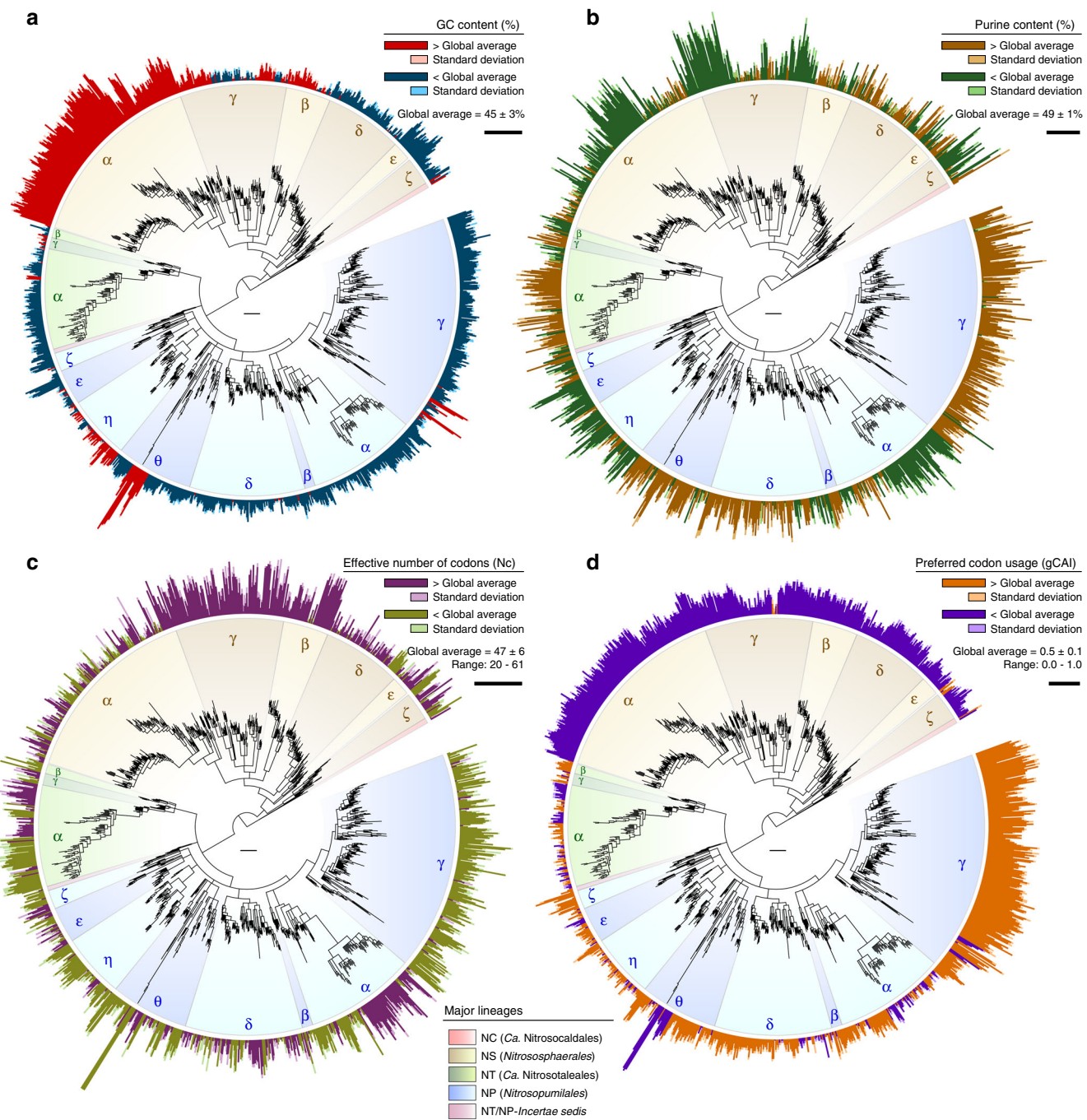

**Fig. 8** Phylogenetic distribution of variation in DNA base composition and synonymous codon usage of archaeal *amoA* genes. **a** Relative GC content (%). Bars represent average GC content variation per OTU in relation to the global average: red, GC percent above average; blue, GC percent below average; the scale bar represents 5% GC. **b** Relative purine content (%). Bars represent average purine content variation in relation to the global average: brown, purine percent above average; green, purine percent below average; the scale bar represents 1% purine. **c** Relative effective number of codons (Nc)[59]. Bars represent average Nc variation in relation to the global average: violet, codon usage more relaxed than average; olive green, codon usage more biased than average; the scale bar represents 10 Nc units. **d** Synonymous codon usage preferences, expressed as the codon adaptation index (gCAI)[62] in relation to the overall codon bias among *amoA* genes; the global gCAI average represents the average deviation from the dominant codon bias. Bars represent average gCAI variation in relation to the global average gCAI: orange, codon preferences more similar to the global preferences than average; purple, codon preferences more distinct from the global preferences than average; the scale bar represents 0.1 gCAI units. Stacked bars in brighter colours represent standard deviations of the average differences in relation to the global average. Bars are drawn in proportion to the highest value according to the respective scales. Values were calculated based on all 23,395 unique sequences in the curated database. Tree scale bars represent 0.1 substitutions per nucleotide position. The phylogenetic/taxonomic distribution of molecular features is also available in an interactive Krona[47] chart in Supplementary Data 10. See legend of Fig. 1 for tree description

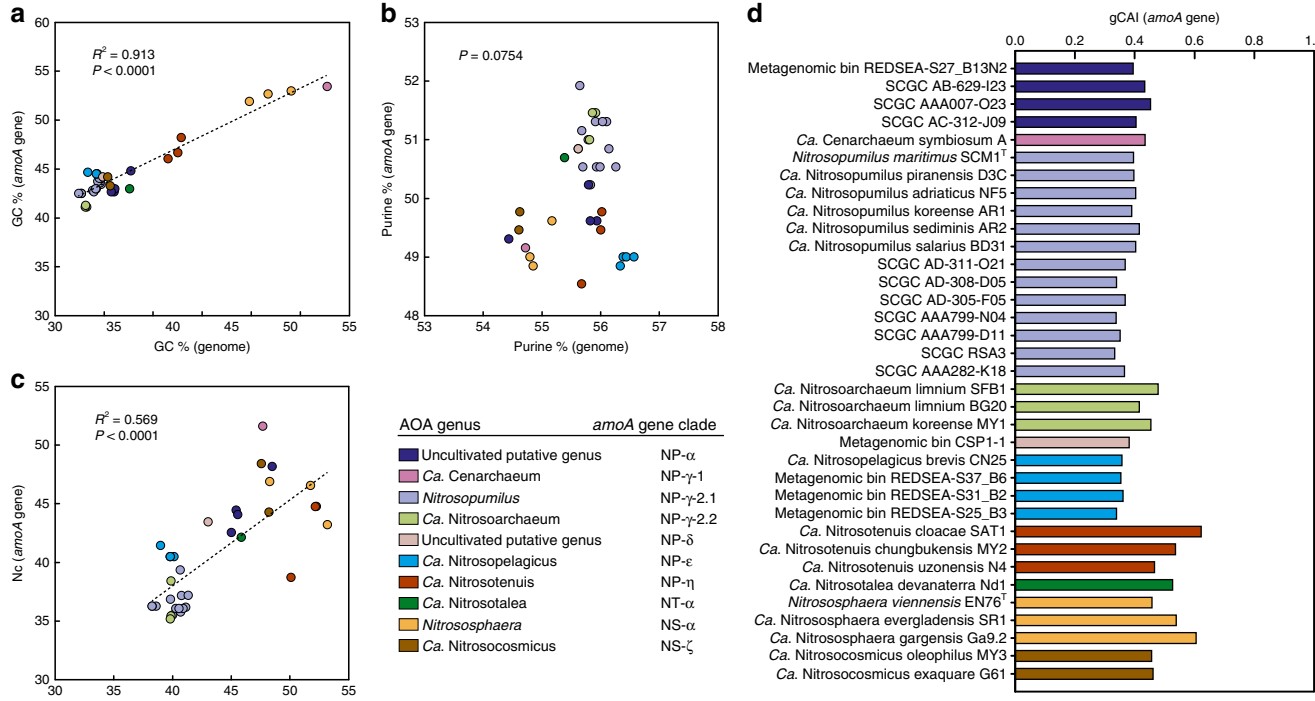

**Fig. 9** Relationship between DNA base composition and synonymous codon usage of *amoA* genes and genomes of AOA. **a–c** Linear regression models of relationships between GC content (%), purine content (%) and effective number of codons (Nc)[59] of full-length *amoA* genes and those of all protein-coding genes in their corresponding genomes. Linear regression lines and coefficients of determination ($R^2$) are shown only for significant relationships, as indicated by their respective *P* values. **d** Global codon adaptation index (gCAI)[62] of *amoA* genes in relation to all available protein-coding genes in the corresponding genomes. Only genomes with >1000 annotated protein-coding genes were included (35 genomes). Filled circles (**a–c**) and bars (**d**) are coloured according to genus, and the corresponding *amoA*-based taxonomy is indicated. Taxonomy and references of organisms and genomes shown are provided in Supplementary Table 1

codon usage of individual genes with that of a reference set of genes by accounting for the dominant codon bias in the latter, and varies between 0 and 1, with higher values indicating codon usage more similar to those of the reference sequences (i.e., global bias)[62]. Variation in gCAI of *amoA* genes shows a conspicuous separation between codon usage of lineages NS and NP, with genes from the latter using more globally preferred codons, and with clade NP-γ showing the highest similarity with the global codon bias (Fig. 8d, Supplementary Data 10). Surprisingly, gCAI values of *amoA* genes calculated in relation to their respective genomes show that *amoA* genes use codons that differ extensively from the genomic codon bias, contrary to the expectation that a gene essential for growth would use globally preferred codons (Fig. 8d). Although the genomic gCAI of *amoA* genes also varies considerably between organisms, the relatively low values indirectly indicate that these differences do not mirror the variation in codon usage among *amoA* genes alone.

**Evolutionary origin of the archaeal AMO within CuMMOs.** To infer the possible evolutionary origins of the archaeal AMO, we analysed the phylogeny of the CuMMO superfamily, which also includes bacterial ammonia and hydrocarbon monooxygenases. Codon-based phylogenetic analysis of 125 concatenated subunits A and B from all known CuMMOs resolved robust relationships between four main lineages largely consistent with phyla, regardless of their known substrates: Thaumarchaeota; Actinobacteria and an apparent lateral acquisition by Deltaproteobacteria; Verrucomicrobia; and Proteobacteria, including AMOs of comammox Nitrospirae, likely acquired from the former (Fig. 10). The CuMMO from Candidate division NC10 was

suggestively placed at the root of proteobacterial CuMMOs, although with low branch support. Regardless of the evolutionary pathway, most possible scenarios indicated that archaeal AMOs share a more recent evolutionary history with actinobacterial monooxygenases than with those of other Bacteria, including bacterial AMOs (see also Supplementary Discussion).

## Discussion

Studies of functional marker genes, such as *amoA*, greatly rely on variable custom phylogenies to infer and interpret diversity patterns, given the general unavailability of reference phylogenies and taxonomies. However, reconstruction of reliable phylogenetic trees of archaeal *amoA* genes, in particular, is notoriously difficult, as evident from many largely unsupported and extensively multifurcated or ladder-like trees[23,24,29,31,32]. By exhaustively accounting for artefacts and stochastic tree variability, we reconstructed a robust archaeal *amoA* gene phylogeny and defined a novel multilevel taxonomy. In addition to the updated gene diversity included, the resolution of our phylogeny/taxonomy contrasts strongly with that of previous large archaeal *amoA* phylogenies, which could only resolve order-level lineages and sparse subclades. For example, the only available reference *amoA* phylogeny/taxonomy[63] used in studies of AOA could not distinguish important environmental clades, such as NS-δ (the most frequently detected AOA clade, represented by fosmid clone 54d9), and fundamental phylogenetic relationships, such as the monophyly of genera *Nitrosopumilus* and *Ca.* Nitrosoarchaeum (two of the best studied genera). Importantly, we detected a large number of chimeras among publicly available *amoA* sequences, which particularly deteriorate tree topology and likely constituted

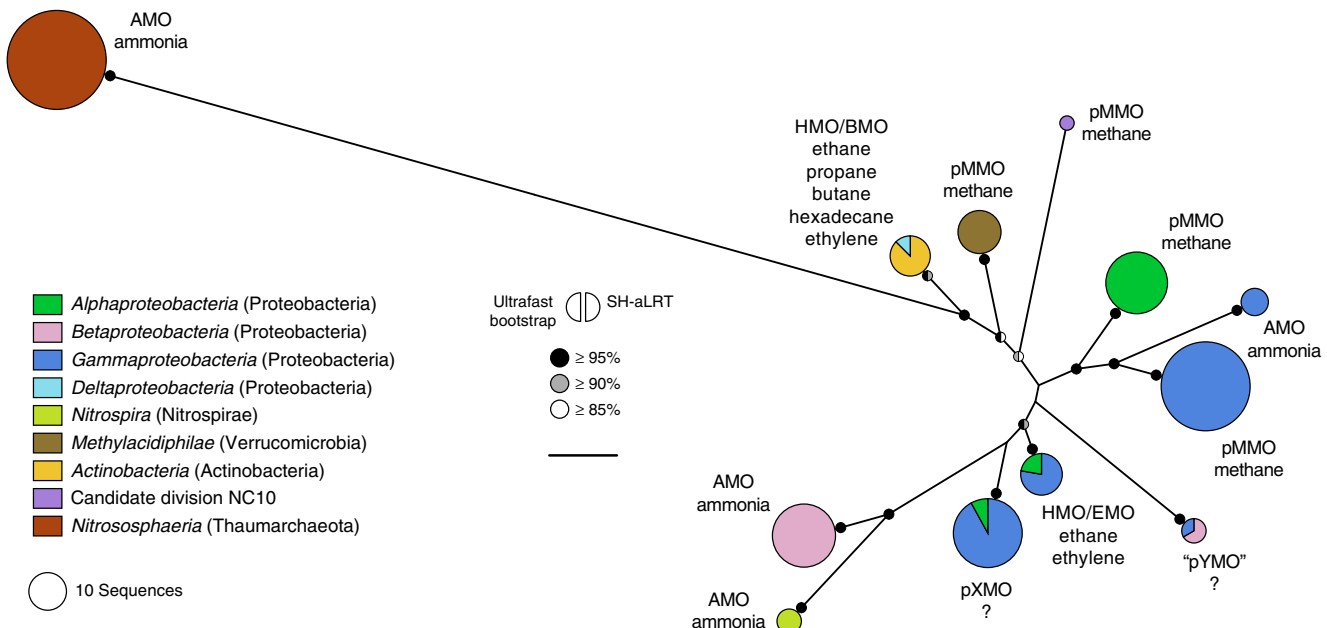

**Fig. 10** Phylogenetic placement of the archaeal AMO within the CuMMO superfamily. Phylogeny of the CuMMO superfamily inferred from 125 concatenated subunits A and B (595 aligned codon positions), representing 196 available sequences clustered at 98% protein identity. Clades are collapsed as circles/pie-charts coloured according to taxonomic class (phyla are indicated in brackets); circle diameter is proportional to number of sequences before clustering. Enzyme names and respective substrates known to support growth are indicated next to clades. "pYMO" was identified here and provisionally named as an additional clade of enzymes with unknown substrate, phylogenetically distinct from pXMO[19]. The tree was inferred by maximum likelihood with IQ-TREE[80] using an edge-proportional partition model with codon models SCHN05+F+Γ4 and KOSI07+F+R4 for subunits A and B, respectively. Ultrafast bootstrap[37] and SH-aLRT[38] support values ≥85% (1000 replicates) are indicated by semi-circles. The scale bar (below branch support legend) represents 0.5 substitutions per codon position

a critical factor underlying the low resolution of other phylogenies. Consistent with this, we found that the early database/phylogeny by Pester et al.[63] contains at least 25% chimeras.

The lack of robust phylogenetic frameworks to classify and integrate biological patterns of archaeal *amoA* genes has largely precluded accurate comparisons and generalisations across studies. While the diversity, abundance and activity of AOA have been related to several factors, the lineages underlying these patterns have remained mostly anonymous or difficult to identify[23,24,64]. In turn, the subjective identification of clades associated with recurrent patterns has led to phylogenetically incongruent ecotypes between studies, as illustrated by the association of different AOA clades with distinct ocean water depths. As previously observed, different ecosystems tend to harbour distinct AOA, which show considerable habitat specificity[5,24,27,64]. However, we show that the global phylogenetic scale and degree of specificity of these patterns vary much more than typically assumed. Besides the broad segregation of AOA between marine and terrestrial ecosystems, their diversification reflects diverse niche adaptation events, including clades with vast ecotypical intra-diversity, and also distantly related clades independently adapted to similar environments. Some ecosystems harbour very specialised ecotypes, namely seawater, marine symbionts and acidic soils, which likely represent organisms with very specific ecophysiological adaptations. Conversely, other clades represent remarkably versatile organisms, such as those widespread in non-acidic soils and different aquatic environments, including freshwater and estuarine–coastal ecosystems. Our study also reveals that the global detection frequency of AOA is extremely uneven across their phylogenetic diversity, with few clades overwhelmingly dominating overall AOA diversity in most individual ecosystems. Moreover, we show that current knowledge about AOA biology is limited to strains that do not

necessarily represent the predominant organisms in nature, including globally relevant ecosystems, namely soils and oceans. Given the functional heterogeneity among cultivated AOA (including closely related strains[65]) and evidence that growth of some AOA might not depend on nitrification[45,46], the overlapping phylogenetic bias of cultivation and SIP-based studies possibly reflects very fundamental physiologic and metabolic discrepancies between clades. This could have also contributed to the frequent lack of correlation between nitrification activity and AOA abundance (in contrast to that of AOB)[45,46,66,67], which is typically just ascribed to sub-optimal environmental conditions. While the global patterns observed synthesise the knowledge that can be inferred from the *amoA*-based information currently available, some diversity might have been systematically undetected due to PCR primer biases (see Methods). Nevertheless, the fact that all *amoA* genes identified through PCR-independent approaches belong to clades widely represented by PCR-amplified genes suggests otherwise (Supplementary Table 1). It is also possible that further studies of currently under-sampled environmental niches, such as biofilms, eukaryotic hosts and subsurface environments, might reveal novel AOA diversity and distribution patterns.

Nucleobase composition and synonymous codon usage affect nucleic acid stability and gene expression, and variation in these properties can result from neutral processes or selection on DNA, RNA or protein levels related to an organism's lifestyle[58,60,61,68–72]. For example, codon usage and GC content of archaea and bacteria have been associated with habitat[58,60,61,69,71] and translational fine-tuning[70], while GC content, in particular, has also been related to specific phenotypic properties, namely growth rate[60,68] and aerobicity[58,69]. Remarkably, *amoA* genes exhibit distinct signatures of GC and purine contents, and codon usage, which are congruent with clades on variable phylogenetic

levels and, in some cases, suggestively associated with their environmental occurrence. Moreover, distinct *amoA* gene signatures reflect genome composition to different extent, and thus differences among *amoA* clades likely reflect trade-offs between genome composition and selection acting specifically on *amoA* genes, transcripts and/or proteins. The variability between codon usage of *amoA* genes and their respective genomes particularly indicates that *amoA* codons have been differentially selected between organisms, independently of their genomic preferences. These results suggest that different AOA clades favour particular amino acids and/or use distinct strategies to regulate *amoA* gene expression, possibly through specific messenger RNA structures or usage of codons matching specific transfer RNA pools. Although the biological significance of associations between molecular signatures and habitat of AOA is unclear, these suggest that niche adaptation directly or indirectly contributed to selection of specific *amoA* gene traits. Furthermore, these signatures independently support the phylogenetic coherence of several clades, and provide additional criteria to differentiate AOA diversity (e.g., as putative phenotypes), independently of evolutionary relatedness or environmental distribution.

The phylogenetic placement of archaeal AMOs within CuMMOs has remained largely ambiguous, as previous phylogenies of individual CuMMO subunits could not reproducibly resolve basal relationships between lineages and often did not consider all the enzyme diversity currently known[19,21]. Based on the different possible scenarios inferred from a codon-based phylogeny of known CuMMOs, we hypothesise that contemporary archaeal AMOs and bacterial monooxygenases evolved as two independent lineages from a common ancestral bacterial or archaeal enzyme. This is supported by the disproportionally greater evolutionary divergence of archaeal AMOs in relation to bacterial monooxygenases than among the latter, and by the central metabolic role of AMO in all organisms of class *Nitrososphaeria*; by contrast, CuMMOs are only present in a small fraction of bacterial taxa within their phyla or classes, which suggests a more recent enzyme acquisition from an ancestral bacterial monooxygenase, followed by diversification on lower taxonomic levels (see also Supplementary Discussion).

Regardless of thorough data selection and analysis, it should be emphasised that phylogenies represent hypotheses about evolutionary relationships between genes, proteins and organisms, whose validity and biological significance should be tested. Nevertheless, the present phylogeny-informed *amoA* taxonomy provides an essential system to reproducibly classify and compare AOA diversity across multiple levels, which is not constrained by arbitrary sequence identity thresholds, or subjective associations with habitats or cultivated organisms. Additionally, by explicitly integrating environmental information and molecular signatures of AOA with this taxonomy, we provide a comprehensive framework to interpret and test hypotheses about AOA ecology and activity in a global context.

## Methods

**Assembly and alignment of archaeal *amoA* sequences**. Archaeal *amoA* sequences were collected from GenBank[73] on 11 November 2014 using the search terms "ammonia monooxygenase subunit A", "ammonia monooxygenase alpha subunit", and amoA[gene], together with "archaea". Unannotated metagenomic sequences were collected through BLASTN[74] searches using sequences from representative strains as query. Over 99% of the 45,136 sequences collected were generated by PCR, mostly using two similar primer pairs that amplify 592–595 bp gene fragments, excluding primers[22,75]. Sequences containing ambiguous bases or labelled "unverified" in GenBank[73] were excluded. Sequences were screened in both orientations with *getorf* (EMBOSS[76]) to extract in-frame sequence fragments with ≥582 bp (194 codons) and to trim partial codons from sequence termini. This length threshold was selected to keep sequences with up to three potential biological codon deletions, given the intended final sequence length of 591 bp. This procedure also removed sequences

containing incorrect stop codons due to sequencing or PCR errors (e.g., frameshifts) in the region of interest. Sequences containing frameshifts were further filtered out using FrameBot[77]. To facilitate sequence alignment analyses, the dataset was transiently de-replicated using mothur v1.33[41]. Selected sequences were aligned based on translated protein sequences with MAFFT v7[78] (FFT-NS-2 method), followed by manual alignment inspection and exclusion of unspecific sequences wrongly annotated as archaeal *amoA* genes, and sequences containing previously undetected frameshifts. Sequences were re-aligned as described above and the alignment was back-translated using *tranalign* (EMBOSS[76]). Redundant sequences excluded by previous de-replication were introduced into the alignment with the "seed alignment" option in MAFFT v7[78]. The filtered aligned dataset containing 35,372 sequences was globally trimmed to 591 bp, representing the longest coding region common to most sequences, and shorter sequences were excluded.

**Chimera filtering and clustering of archaeal *amoA* sequences**. Screening for chimeric sequences was performed with UCHIME[34], ChimeraSlayer[33] and by intensively inspecting phylogenetic trees and sequence alignments. Chimera screening with UCHIME[34] and ChimeraSlayer[33] was based on a newly assembled reference database, since the different dataset sizes and preparation protocols of the studies included were not compatible with assumptions of efficient de novo approaches[36]. Assembly of the reference database followed the principle that more abundant sequences are more likely to be true biological sequences; since sequence libraries generated by Sanger sequencing, as those analysed here, have generally low sequencing depth, the likelihood of detecting the same chimera multiple times is extremely low, particularly across independent datasets. Therefore, the reference database was initially compiled by selecting sequences occurring at least five times in the assembled dataset (35,372 sequences) using USEARCH v8.1[79]. This threshold was selected after observation of multiple depositions of the same sequences in GenBank[73] under different accession numbers, including chimeras, as evident from the studies' authors and titles. Additional sequences were selected based on a preliminary phylogenetic tree to improve the coverage and evenness of the reference database[34]. The tree was inferred by maximum likelihood with IQ-TREE[80] using the best-fit model of DNA evolution, based on 2203 OTU representatives of the unfiltered dataset (35,372 sequences) clustered at 96% sequence identity with the average-neighbour method in mothur v1.37[41]. All sequences were inspected based on BLASTN[74] searches against GenBank[73] and were included in the databases only if: (i) they shared ≥99% full-length identity with at least two other sequences, including at least one from an independent study; (ii) no obvious chimera could be detected based on their alignments with multiple sequence hits. Some exceptions to these criteria were made to improve phylogenetic coverage, by allowing BLASTN[74] hits with ≥98% sequence identity and exhaustively inspecting the respective alignments. The database was complemented with sequences from available genomes and then clustered at 97% sequence identity, as described above. UCHIME[34] parameters were optimised for archaeal *amoA* genes using our reference database, as follows: (i) the main parameters -*mindiv* (minimum score cut-off) and -*minh* (minimum identity between query and closest reference sequence) were initially adjusted to minimise false positives by screening the reference database using it as both query and reference[34]; (ii) the full *amoA* dataset (35,372 sequences) was screened using several combinations of -*mindiv* and -*minh* values, and a random fraction of chimeras flagged in each run was inspected based on BLASTN[74] searches; sequences that complied with the BLASTN-based criteria described above were considered false positives, and suspected chimeras were further investigated through similar BLASTN[74] analysis of their GenBank[73] hits to verify whether these were chimeras themselves. This procedure was repeated after every parameter adjustment until detection of confirmed chimeras was maximal without increasing false positives, as estimated from all sequences verified in previous iterations. The final UCHIME[34] parameters selected were -*mindiv* 1.7 and -*minh* 0.10. Additional chimeras were identified with ChimeraSlayer[33] (mothur v1.33[41]) using the *blast* approach with a minimum bootstrap (-*minbs*) value of 95%, after testing values of 90 and 93% following the same criteria used for UCHIME[34]. Sequences unambiguously confirmed as false positives during optimisation procedures were kept in the database. The final reference database comprised 411 sequences after re-clustering at 97% identity. Filtering the *amoA* sequence dataset (35,372 sequences) with UCHIME[34] and ChimeraSlayer[33] removed 1532 chimeras, 804 of which were detected by both methods. Fifteen verified false positives flagged in the last chimera screening were re-introduced in the main dataset. Based on analysis of multiple phylogenetic trees, calculated as described above, we identified 462 additional chimeras by inspecting sequences with unusually long and isolated branches, and/or forming extensive ladder-like topologies. This procedure was repeated after every round of sequence inspection and exclusion, until no new chimeras were identified. Only unambiguous chimeras were removed, based on extensive analysis of sequence alignments as described above. A total of 1994 chimeras were removed during all screening procedures; 2361 sequences were manually inspected, including 926 confirmed chimeras, 617 true biological sequences and 818 sequences that could not be confirmed as either. The final curated database comprised 33,378 aligned sequences, including 23,395 unique sequences, represented by 1206 OTUs after re-clustering at 96% sequence identity, as described above. This threshold was selected to minimise data redundancy and spurious diversity derived from sequencing errors in order to facilitate

subsequent phylogenetic analyses, while retaining high gene diversity resolution, namely in relation to thresholds commonly used to define *amoA* OTUs (≥95%), or estimated to represent different species (<87%).

As the database was assembled in November 2014, we collected and curated, as described above, all new 13,026 *amoA* sequences deposited in GenBank[73] as of 17 November 2016 to determine if newly available gene diversity was accounted for. Using USEARCH v8.1[79], we verified that 94.8% of new sequences could be assigned to OTUs in our database at ≥96% sequence identity, whereas 99.6% could be matched at ≥90% identity.

**Phylogenetic analysis and taxonomy of archaeal *amoA* genes.** Sixteen rogue sequences were identified and excluded from further phylogenetic analyses based on inspection of multiple trees inferred with IQ-TREE v1.2.2[80], by iteratively excluding–replacing isolated sequences suspected to affect the topological stability. Rogue sequences represent taxa with insufficient or ambiguous phylogenetic signal (e.g., underrepresented divergent lineages) that lead to tree topology and branch support instability[81]. After rogue sequence exclusion, the alignment comprised 1190 OTU representatives. Selection of the best-fit model of evolution, GTR+F+I +Γ4, was performed with IQ-TREE v1.2.2[80], and phylogenetic trees were inferred by maximum likelihood (ML) with IQ-TREE v1.2.2[80] and PhyML 3.0[38], and Bayesian inference with MrBayes 3.2[82]. IQ-TREE[80] was run independently 10 times with each of 5 perturbation strength values for randomised NNI search, and a stopping rule for tree search of 500 iterations; ultrafast bootstrap (UFBoot)[37] and SH-aLRT[38] support values were calculated from 1000 replicates. The 10 best-known ML trees inferred with the perturbation strength yielding the highest tree log-likelihood values (–*pers* 0.1) were selected for further analysis. PhyML[38] was run independently 10 times with combined SPR and NNI tree search using a BIONJ tree and two random trees as starting trees, and branch support was calculated with SH-aLRT[38]. MrBayes[82] was run independently 10 times with four chains and 50 million MCMC generations. Analysis with Tracer v1.6 (Rambaut, A., Suchard, M., Xie, D., Drummond A. J., http://tree.bio.ed.ac.uk/software/tracer/, 2013) showed that only two of the runs converged, and at much lower log-likelihood values than those obtained with IQ-TREE[80] or PhyML[38]; therefore, MrBayes[82] results were excluded from further analyses. Relative confidence of the 20 best-known ML trees inferred with IQ-TREE[80] and PhyML[38] was analysed based on weighted Kishino–Hasegawa[83] and Shimodaira–Hasegawa[84] tests, expected likelihood weights (ELW)[85] and the approximately unbiased test[86]; tests were performed with IQ-TREE[80] using the best-fit model (GTR+F+I+Γ4) and 1000 bootstrap replicates. The comparative topology tests consistently showed that no single tree explained the data significantly better than the others, except for two trees that failed only the ELW[85] test, and were therefore not excluded. Further tree selection was performed based on congruency with a concatenated 16S and 23S rRNA gene phylogeny (inferred as described below), as an external biological reference, and based on congruency among alternative *amoA* trees. All known AOA harbour a single *amoA* gene copy, and *amoA* and 16S rRNA gene phylogenies are known to be congruent, at least on the order level[6,63]; therefore, tree selection followed the assumption that, among alternative and statistically undistinguishable *amoA* tree topologies, those compatible with the rRNA gene tree are more likely to be correct. Although the *amoA* trees represented many more organisms than the rRNA gene tree, several clades ranging from genus to order could be linked between trees based on available genomes. Among the 20 *amoA* trees analysed, 9 independent trees inferred with IQ-TREE[80] were congruent with the rRNA gene tree, or compatible on the basis that topological differences could have resulted from the absence of broad intermediate lineages in the latter. Congruency among the nine selected trees was analysed with RAxML v8.1.21[87] based on internode certainty scores (IC) of each tree, estimated from internode frequencies across all nine trees. The tree with the highest tree certainty score (sum of all IC scores) was selected as the best tree—that is, the tree with the highest overall congruency with all other eight trees. Branches of the best tree with ultrafast bootstrap[37] <95% and/or SH-aLRT[38] <85%, each threshold corresponding to an estimated confidence level of 95%[37], were collapsed, and remaining branch lengths were re-calculated with the GTR+F+I+Γ4 model. Rogue sequences excluded from phylogenetic analysis were re-introduced in the database and classified using the Evolutionary Placement Algorithm[44].

Taxonomic ranks were defined strictly based on supported branches according to the criteria above. Ranks represented by Greek letters or numbers were ordered starting from the clade with the longest branch among those in the same previous rank. Taxonomic ranks were only assigned to clusters of at least three OTU representatives, and clusters of two sequences were designated *incertae sedis* (abbreviated to "-IS") to indicate their monophyly but uncertain status as separate clades. To minimise redundancy, ranks were only assigned when necessary to distinguish between at least two clades sharing the same previous rank (e.g., NP-α-1.1 and NP-α-1.2); consequently, clades without at least two clusters of at least three sequences represented terminal ranks. Four specific exceptions were made to this criterion, by assigning new ranks to clusters with long branches that represented conspicuously distinct subclades in relation to other sequences in the same previous rank (NP-α-2.2.2.1, NP-θ-4.1.1, NT-α-1 and NC- α). It should be noted that the taxonomy strictly reflects phylogenetic relationships, and thus distinct (sub)clades with the same rank level (e.g., NP-α-1 and NP-α-2) are not necessarily equivalent.

**Analysis of environmental and SIP-based meta-data.** Environmental source information of all *amoA* sequences in the curated database (33,378 sequences) was parsed from GenBank[73] files using a custom Python script (Supplementary Data 11). If information was missing or described in ambiguous terms, it was collected from the original publications when available. Given the diversity of terms and level of detail of sequence source annotations, we categorised sequences manually into unambiguous environmental categories on different levels. To minimise local- or study-specific biases, we characterised only environmental categories to which we could assign a substantial number of sequences from different studies (>200 sequences, with the exception of corals, but from several hundred to >1000 in most cases). Consequently, many specific environmental niches were not sufficiently represented in the data to infer robust global patterns. Categories were defined as to maximise sequence inclusion while avoiding ambiguities. Sequences were categorised as marine only when detected in the water column or seafloor of oceanic regions, as well as salt water aquaria. Few sequences collected from seafloor precipitates in hydrothermal vent fields were grouped with those from marine/oceanic sediments. To minimise ambiguous categorisations, sequences found in coastal areas (e.g., bays) were grouped with those from other marine–terrestrial interfaces, namely estuaries and coastal wetlands (i.e., marshes and mangroves), as they are similarly under continuous influence of terrestrial ecosystems. Therefore, sequences categorised as estuarine–coastal likely include some true marine and terrestrial sequences. Estuarine–coastal sequences were not divided between water and sediments, since these could not always be clearly distinguished based on the information available (e.g., in shallow areas). Likewise, sediments from freshwater wetlands, such as riparian and flooded areas (e.g., paddies), were grouped with soils, as they could not be systematically differentiated. Sequences categorised as "acidic-neutral" or "neutral-alkaline" represent those for which pH information was reported as a range spanning different pH categories in the original study, e.g., the "acidic-neutral" category represents sequences that might derive from either acidic or neutral soils, according to the definitions used here. Sequence counts for each habitat were mapped to the 1206 OTUs in the database using mothur v1.37[41].

Sequence information from archaeal *amoA*-based SIP experiments could be collected from 12 of the studies found in PubMed (Supplementary Table 2). Representative *amoA* sequences and their relative abundances in $^{13}$C-labelled SIP (i.e., heavy) fractions were obtained from the original publications, namely from phylogenetic tree displays. In case sequences could not be found in GenBank[73] based on the identifiers provided, their closest phylogenetic representatives were used, when available. Forty-four representative sequences were assigned to OTUs in the database at ≥96% identity using USEARCH v8.1[79].

**Analysis of nucleobase composition and codon usage.** GC and purine contents (%) of all unique *amoA* genes in the curated database (25,231 sequences) were calculated with MEGA6[88] and the effective Nc[59] was calculated with *chips* (EMBOSS[76]). To compare codon preferences among *amoA* genes, the gCAI[62] was calculated for all unique sequences with JCat[89], using all OTU sequence representatives (1206 sequences) as reference. Variation in GC percent, purine percent, Nc and gCAI was expressed as the average difference (i.e., arithmetic mean ± standard deviation) between unique sequences in each of the 1206 OTUs and the global average; that is, the average OTU deviation from the average of all unique *amoA* genes. In the case of gCAI, the global average represents the average deviation from the dominant codon usage bias, with higher values indicating codon usage more similar to the global usage among all *amoA* genes. To avoid biases due to different OTU sizes, all global average values were calculated as the average of all OTU average values.

Genomic GC percent, purine percent and Nc were calculated from all protein-coding gene sequences in archaeal genomes containing *amoA* genes. Sequences were collected from IMG/M[90] and included genomes of cultivated organisms, single-cell-amplified genomes and assembled metagenomic bins. Thirty-five genomes containing 1194–3722 protein-coding genes were included, and partial genomes with <1000 protein-coding genes were excluded to minimise biases in further analyses. gCAI of *amoA* genes in relation to their genomes was calculated by comparing the codon usage of each full-length *amoA* gene with that of all protein-coding genes in the respective genome. Relationships between GC percent, purine percent and Nc values of full-length *amoA* genes and those of their respective genomes were analysed using linear regression models with SigmaPlot v11.0 (Systat Software Inc., San Jose, CA, USA; www.sigmaplot.com). Assumptions of linear regression were tested using the Shapiro–Wilk test (normality of residuals) and residual plots (equal variance) with Statgraphics 5.0 (Statistical Graphics Inc., Rockville, MD, USA; www.statgraphics.com).

**Phylogenetic analysis of rRNA genes.** Full-length 16S and 23S rRNA gene sequences from available genomes of Thaumarchaeota and candidate taxa in the TACK superphylum ($n = 112$), as well as representatives of all Crenarchaeota genera ($n = 22$) and 8 Euryarchaeota classes (outgroup), were collected from GenBank[73] and IMG/M[90]. If only partial gene sequences were annotated, the complete sequences were extracted based on BLASTN[74] searches and alignments with reference sequences. Only sequences from genomes containing both full rRNA genes were included. Sequences were aligned before concatenation with MAFFT v7[78] (L-INS-I method) based on structurally accurate seed alignments of

archaeal 16S and 23S rRNA gene sequences obtained from the Comparative RNA website on February 2015[91]. To preserve structural accuracy, alignments were manually corrected for column shifts caused by long indels in new sequences from taxa underrepresented in the seed alignment. Concatenated 16S and 23S rRNA gene sequences were clustered at 99% identity with the average-neighbour method in mothur v1.37[41] and alignment columns with >90% gaps were excluded. The final concatenated alignment comprised 92 sequences with 4628 aligned positions. The best-fit model of evolution was selected with ModelFinder[92] and phylogenetic trees were inferred by ML with IQ-TREE v1.5.3[80] based on the model GTR+F+R6, with parameters estimated separately for each gene using an edge-unlinked partition model[93]. The tree was inferred independently 10 times and the best-known ML tree with the highest log-likelihood was selected. Ultrafast bootstrap[37] and SH-aLRT[38] support values were calculated from 1000 replicates with IQ-TREE[80].

**Phylogenetic analysis of CuMMO proteins.** Gene sequences encoding subunits A and B of CuMMO enzymes were collected from GenBank[73] and IMG/M[90] through BLASTP[74] searches using reference sequences from known lineages as query; in cases of partial gene sequence annotation, complete sequences were extracted as described for rRNA genes. Only sequences from genomes containing complete genes for both subunits were included (196 protein pairs), with the exception of subunit B of Ca. N. yellowstonensis HL72, which lacked 4 codons. Concatenated protein sequences of subunits A and B were clustered at 98% identity with UCLUST[79]; in case multiple non-identical operons were present in the same genome, correct subunit pairs were concatenated based on their genomic coordinates. Given the very low similarity and different length among proteins, sequences of each subunit were separately aligned with 14 different methods: L-INS-i, E-INS-i and G-INS-i strategies of MAFFT v7[78]; PSI-Coffee[94], EXPRESSO[95]; T-Coffee[96]; ProbCons v1.12[97]; Probalign v1.3[98]; Prank[99]; Muscle v3.7[100]; MSAProbs v0.9.7[101]; Kalign[102]; Dialign-TX[103]; and Clustal Omega[104]. The 10 best alignments were selected after evaluation with MUMSA[105] based on multiple overlap (MOS) and average overlap (AOS) scores. Alignments with MOS above the average of the 10 alignments selected were merged with MergeAlign[106], and columns not supported by at least two independent alignments, or containing >80% gaps, were excluded. The consensus alignment was based on PSI-Coffee[94], MSAProbs[101], G-INS-i, E-INS-i and L-INS-I for subunit A, and on G-INS-i, E-INS-i, MSAProbs[101] and L-INS-I for subunit B. The final alignment of concatenated CuMMO subunits A and B comprised 125 sequences with 595 aligned protein/codon positions. Analysis with ModelOMatic[107] indicated that codon models fit the data better than protein models and thus phylogenetic analyses were performed based on codon sequences. The best-fit model of codon evolution was selected with ModelFinder[92] and phylogenetic trees were inferred by ML with IQ-TREE v1.5.5[80] using an edge-proportional partition model[93] with models SCHN05+F+Γ4 and KOSI07+F+R4 for subunits A and B, respectively. The tree was inferred independently 10 times and the best-known ML tree with the highest log-likelihood was selected. Ultrafast boostrap[37] and SH-aLRT[38] support values were calculated from 1000 replicates with IQ-TREE[80].

**General data analysis and visualisation.** All manipulations of sequence files were performed with mothur[41] and USEARCH v8.1[79]. Phylogenetic trees were visualised with FigTree (Rambaut, A., FigTree v1.4.3, http://beast.bio.ed.ac.uk/figtree, 2016), and environmental and molecular composition data were displayed in the phylogeny with iTOL v3[108], and in hierarchical charts using Krona[47]. MAFFT[78], Muscle[100], ProbCons[97] and Probalign[98] analyses were performed through the CIPRES Science Gateway[109], and EMBOSS[76] tools were used through the Galaxy platform[110] (http://usegalaxy.org).

**Code availability.** Custom Python code used in this study is provided with this article in Supplementary Data 11.

**Data availability.** All primary data used during this study are publicly available from GenBank[73] or IMG/M[90], as indicated in the Methods section and Supplementary Tables 1 and 2. Curated datasets and reference databases and trees compiled/analysed during this study are provided with this article in Supplementary Data 1-3.

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

## Acknowledgements

We thank Sophie Abby for help with bioinformatic tools and helpful discussions, Maria Mooshammer for help with statistical analyses and comments on the manuscript, and Anders Lanzén for compiling the CREST and MEGAN databases. We also thank Pierre Offre for insightful discussions and Brian Ondov for prompt technical assistance with the Krona software. This work was funded by project P25369 of the Austrian Science Fund (FWF) and ERC Advanced Grant project TACKLE (No 695192, awarded to C.S.); T.U. acknowledges financial support from ESF and Ministry of Education, Science and Culture of Mecklenburg-Western Pomerania project WETSCAPES (ESF/14-BM-A55-0032/16).

## Author contributions

R.J.E.A., T.U. and C.S. conceived the project; R.J.E.A. and B.Q.M. performed phylogenetic analyses; R.J.E.A. assembled, curated and analysed sequence data and associated meta-data and, together with T.U. and C.S., interpreted the results; A.v.H. and C.S. provided computational resources; R.J.E.A. wrote the manuscript with contribution from C.S.; all authors discussed and revised the manuscript.

## Additional information

**Competing interests:** The authors declare no competing interests.

