## [Peer Review File · Nature Communications]

Reviewers' comments:

Reviewer #1 (Remarks to the Author):

The manuscript by Alves et al. reports original insights into the phylogeny, diversity, and distribution of the ecologically important ammonia-oxidizing archaea based on the extensive analysis of amoA gene sequences from current databases.

This is a generous and extremely well done study, which likely took a few years to complete.

It has happened only once previously in my activity as reviewer, but I have to admit that do not see any particular problem with this paper, which in my opinion can be published as is. It is clearly written, the amount of data analysed is enormous, and the results are thoroughly presented and convincingly discussed. The quality of figures is remarkable, and the availability by the authors of an interactive phylogenetic tree of amoA sequences as well as a curated database free of chimeras represent an incredibly important tool for the scientific community.

As such, I expect that this paper will become an important reference for the whole community working on this key group of organisms.

I would just suggest being a bit more specific about the main results in the abstract, which presently does not make justice of the importance of this work.

Reviewer #2 (Remarks to the Author):

. The manuscript by Alves and colleagues nicely addresses a timely topic of high interest in current microbial soil and aquatic ecology and in archaeal biology. After extensive amoA gene hunting in different environments for many years and by many different scientists, this work brings the desirable and largely awaited integrative exercise of an extremely curated synthesis of the available information in a very comprehensive manner. The level of detail provided is very high and this work is called to articulate the future research in AOA biology and ecology. The paper is certainly welcome and will be a keystone contribution in the field.

. As a major contribution the paper provides a convincing highly-resolved taxonomy for AOA that will help to understand the ecology, taxonomy and global distribution patterns of archaeal ammonia oxidizers. Interestingly, the most abundant clade lacks cultivated representatives showing that we are still far from a correct understanding of the biology of this group. The work also provides further evidence to support some AOA species are true symbionts of marine sponges and macroalgae.

. Some concerns and clarifications should be however addressed, mainly those referred to the conclusion that only a few clades dominate AOA diversity in most ecosystems, as well as misidentification and undersampling, as follows:

-(i) I miss some sentences to justify why the archaeal amoA gene OTUs were clustered at

96% sequence identity for phylogeny reconstruction. The authors found good agreement with the 16S gene phylogeny but still former papers said that the species identity level that could make sense for amo A gene sequence is far below 96%.

(ii) The priming bias and how good is the current view on the real AOA diversity is also a key point that deserves further discussion. As the authors mention, 99% of the sequences analyzed here were generated by PCR, mostly using similar primer pairs in the different papers. I guess these primers were those designed 10 years ago from a few selected sequences without substantial modifications. According to the long experience that microbial ecologists have with 16S priming biases, most probably we are missing still part of the picture. Then the conclusion that only a few clades dominate AOA diversity in most ecosystems could be turned as an unwarranted assumption. Do the authors agree with that? Could they support the goodness of the primers largely used in the literature according to recent AOA sequences not obtained by PCR methods?

(iii) Current environmental undersampling could also change a bit the picture provided by this paper. For instance, I am missing biofilms data here, and recently we have known that AOA are common in riverine biofilms. That may happen with additional habitats. Could the authors tell us which type of environments should be further explored? It will be also interesting to add in the discussion the recently reported photoinhibition in AOA that could have shaped evolutionary changes.

L. 52 I have checked this reference (number 23) and it deals with temporal patterns in surface waters and not depth-dependent patterns are reported there. Probably the authors wanted to cite Restrepo et al in *Environ Microbiol* 2014 (Targeting spatiotemporal dynamics and segregation of ammonia-oxidizing thaumarchaeota ecotypes).

Finally, it would be worth to mention by the authors that at the end phylogenetic trees are hypothesis that need to be tested. Certainly this paper will encourage microbial ecologists to do it.

Response to referees

Manuscript NCOMMS-17-30740A

Dear editor,

Please find attached our revised manuscript.

We are happy about the overall positive comments of the reviews and have taken into account all suggested changes/additions. Below, we address point-by-point all of the referees' comments. All changes are highlighted in the revised manuscript with the tracking function of Microsoft Word, including additional minor corrections made to comply with Nature Communications' guidelines, or for better readability and brevity, but without changes in content. Additionally, we performed the following minor changes:

Results: We added a link to the latest version of the software "LCAClassifier" for classification of gene sequences (CREST package), which now supports our *amoA* gene database/taxonomy automatically upon installation, without additional files or configurations required (thanks to Anders Lanzén).

Figure 1, Supplementary Figure 2 and Supplementary Table 1: Corrected the name "*Ca. Nitrosocaldus cavascurus* SCU2" to "*Ca. Nitrosocaldus cavascurensis* SCU2".

Supplementary Information: Divided the supplementary text between the sub-sections "Supplementary Discussion" and "Supplementary Methods".

Supplementary Table 1: Changed the title for clarity, and added the accession numbers of the genome of *Ca. Nitrosocaldus cavascurensis* SCU2 and *amoA* gene of *Ca. Nitrosocosmicus arcticus* Kfb, which are now publicly available.

Supplementary Figure 2: Added the accession number of *Ca. Nitrosocaldus cavascurensis* SCU2's genome after the respective name in the figure.

Supplementary Data: Renamed supplementary data files and provide their descriptions in an additional supplementary PDF file (previously included in the Supplementary Information). Custom Python code used to parse sequence source data from GenBank files is now provided in Supplementary Data 9.

Reviewers' comments:

Reviewer #1 (Remarks to the Author):

The manuscript by Alves et al. reports original insights into the phylogeny, diversity, and distribution of the ecologically important ammonia-oxidizing archaea based on the extensive analysis of *amoA* gene sequences from current databases.

This is a generous and extremely well done study, which likely took a few years to complete. It has happened only once previously in my activity as reviewer, but I have to admit that do not see any particular problem with this paper, which in my opinion can be published as is. It is clearly written, the amount of data analysed is enormous, and the results are thoroughly presented and convincingly discussed. The quality of figures is remarkable, and the availability by the authors of an interactive phylogenetic tree of *amoA* sequences as well as a curated database free of chimeras represent an

incredibly important tool for the scientific community. As such, I expect that this paper will become an important reference for the whole community working on this key group of organisms.

I would just suggest being a bit more specific about the main results in the abstract, which presently does not make justice of the importance of this work.

Reply: We are grateful for the positive comments on our manuscript, which was indeed a long-term effort that took a few years to complete.

Given the multiple aspects addressed by the study and the short format of the abstract, we found it difficult to be more precise about our results. Nevertheless, we have now rearranged a bit the text and added a few sentences to better highlight the major findings.

Reviewer #2 (Remarks to the Author):

. The manuscript by Alves and colleagues nicely addresses a timely topic of high interest in current microbial soil and aquatic ecology and in archaeal biology. After extensive *amoA* gene hunting in different environments for many years and by many different scientists, this work brings the desirable and largely awaited integrative exercise of an extremely curated synthesis of the available information in a very comprehensive manner. The level of detail provided is very high and this work is called to articulate the future research in AOA biology and ecology. The paper is certainly welcome and will be a keystone contribution in the field.

. As a major contribution the paper provides a convincing highly-resolved taxonomy for AOA that will help to understand the ecology, taxonomy and global distribution patterns of archaeal ammonia oxidizers. Interestingly, the most abundant clade lacks cultivated representatives showing that we are still far from a correct understanding of the biology of this group. The work also provides further evidence to support some AOA species are true symbionts of marine sponges and macroalgae.

. Some concerns and clarifications should be however addressed, mainly those referred to the conclusion that only a few clades dominate AOA diversity in most ecosystems, as well as misidentification and undersampling, as follows:

(i) I miss some sentences to justify why the archaeal *amoA* gene OTUs were clustered at 96% sequence identity for phylogeny reconstruction. The authors found good agreement with the 16S gene phylogeny but still former papers said that the species identity level that could make sense for *amoA* gene sequence is far below 96%.

Reply: As requested by the referee, we added now a sentence in the Methods' section briefly explaining the reasoning for the clustering threshold used (Lines 521-524). The threshold used represented a compromise between retaining high diversity resolution, while minimising spurious diversity (i.e., derived from sequencing errors) and data redundancy, to allow more accurate and unbiased phylogenetic analyses. While reducing data redundancy is essential to minimise imbalances in the sampling depth of sequence diversity between lineages, a high diversity coverage is also particularly important to infer statistically robust phylogenetic relationships. This threshold was also intended to provide a slightly greater resolution than that of the threshold typically used to define archaeal *amoA* OTUs ($\geq 95\%$ sequence identity), which is itself arbitrary. While a species-level sequence identity of $\geq 87\%$ has indeed been estimated for *amoA* genes, the biological significance of that, or any other thresholds, is largely unclear, regardless of taxonomic ranks. As in other bacteria and archaea, AOA of the same taxonomic species, are known to often represent distinct ecotypes and/or phenotypes. For example, closely related *Nitrosopumilus* species – which could actually be considered the same species based on 16S rRNA gene identity alone ($\geq 99\%$ identity), and whose *amoA* genes share $\geq 93\%$ identity – have been shown to harbour several distinct ecophysiological and functional properties

regarding, e.g., photosensitivity, motility/chemotaxis, urea usage, and adaptation to different temperatures, pH and salinity (Qin *et al.*, 2014, PNAS; Bayer *et al.*, 2015, ISMEJ). As discussed in the manuscript, our results also show that the phylogenetic depth of different AOA ecotypes is variable, and particularly emphasise the importance of analysing biological patterns of AOA on a finer scale – for example, the proposed species-level identity threshold of $\geq 87\%$ would have obscured the hot spring ecotypes and most of the multiple small-scale ecotypes within clade NP-Gamma.

(ii) The priming bias and how good is the current view on the real AOA diversity is also a key point that deserves further discussion. As the authors mention, 99% of the sequences analyzed here were generated by PCR, mostly using similar primer pairs in the different papers. I guess these primers were those designed 10 years ago from a few selected sequences without substantial modifications. According to the long experience that microbial ecologists have with 16S priming biases, most probably we are missing still part of the picture. Then the conclusion that only a few clades dominate AOA diversity in most ecosystems could be turn as an unwarranted assumption. Do the authors agree with that? Could they support the goodness of the primers largely used in the literature according to recent AOA sequences not obtained by PCR methods?

Reply: We agree that this is an important aspect that deserves further discussion. Since most *amoA* sequences available were generated by PCR using mainly the same two primer pairs, it is indeed possible that some diversity might have been systematically undetected, and/or that relative detection of *amoA* genes might have been biased. We have now included some sentences in the Discussion addressing this issue, following the referee's comments (Lines 393-397). Nevertheless, we believe that it is valid to state that only a few clades dominate AOA diversity in most ecosystems in light of current evidence, as inferred from our meta-analysis of the *amoA*-based information currently available. On the other hand, the fact that most sequences were obtained with the same primers can also be seen as a normalising factor for the global patterns inferred, as most studies included here were subject to similar primer biases. We would also like to point out that the large frequency discrepancies among very closely-related *amoA* genes (thus with similar primer affinities) and the fact that this occurs within multiple distantly-related lineages (more likely to have different primer affinities), support that this variation in gene frequency does not derive mainly from primer biases, especially since these patterns reflect the combined information from multiple independent studies. Moreover, as suggested by the referee, all available *amoA* genes obtained through PCR-independent approaches, including cultivated strains, and environmental metagenomes and single-cell amplified genomes (included in our analyses), indeed belong to clades widely represented by PCR-amplified genes (see Supplementary Table 1).

(iii) Current environmental undersampling could also change a bit the picture provided by this paper. For instance, I am missing biofilms data here, and recently we have known that AOA are common in riverine biofilms. That may happen with additional habitats. Could the authors tell us which type of environments should be further explored? It will be also interesting to add in the discussion the recently reported photoinhibition in AOA that could have shaped evolutionary changes.

Reply: It is true that the sampling depth of *amoA* genes varies across environments, although our analyses were necessarily constrained by the data currently available and thus reflect the environmental sampling trends of the past 10 years. Although the patterns observed are supported by large amounts of carefully curated data from independent sources, this meta-analysis can only represent the information currently available. Moreover, the great variability of terms and level of detail used in sequence source annotations also limited the diversity of environmental categories that could be reliably defined, as described in the Supplementary Information. Moreover, in order to infer globally robust patterns with minimal local- or study-specific biases, we only characterised environmental categories to which we could

unambiguously assign a substantial number of sequences from different studies (>200 sequences, with the exception of corals, but from several hundred to >1,000 in most cases). Consequently, many specific environmental niches were not sufficiently represented in the data to infer robust global patterns (this information was now included in the description of environmental category definitions in the Supplementary Methods). This was the case for biofilms, as well as several other habitats where AOA are understudied, such as plant phyllosphere, eukaryotic hosts and subsurface environments. As pointed out by the referee, it is possible that novel patterns and diversity might emerge, as further data from undersampled environments become available. We added now a sentence in the Discussion section (Lines 397-399) emphasising this aspect.

While photosensitivity and other ecophysiological/metabolic properties (e.g., chemotaxis) are important factors in AOA ecology, information linking them with specific clades is very sparse and unclear. For instance, some *Nitrosopumilus* species harbour UV resistance genes (like the distantly related marine AOA *Ca. N. brevis*), while others do not; in turn, the phylogenetic distribution of such properties in other lineages is even more limited. Assessing the role of these properties in AOA evolution/function will necessarily require further genomic, activity and environmental studies, and was outside the scope of our study.

L. 52 I have checked this reference (number 23) and it deals with temporal patterns in surface waters and not depth-dependent patterns are reported there. Probably the authors wanted to cite Restrepo et al in Environ Microbiol 2014 (Targeting spatiotemporal dynamics and segregation of ammonia-oxidizing thaumarchaeota ecotypes).

Reply: We thank the referee for pointing this out. This was a mistake on our part, which happened while streamlining the text. We have now replaced the previous reference number 23 (Auguet et al., 2011) by the correct reference indicated by the referee (Restrepo-Ortiz *et al.* 2014, Environ Microbiol; reference 27).

Finally, it would be worth to mention by the authors that at the end phylogenetic trees are hypothesis that need to be tested. Certainly this paper will encourage microbial ecologists to do it.

Reply: We agree and we have added now a sentence pointing out this aspect in the Discussion as follows: “Regardless of thorough data selection and analysis, phylogenies nevertheless represent hypotheses about evolutionary relationships between genes, proteins and organisms, whose validity and biological significance should be tested.”